# TetraJet-v2: Accurate NVFP4 Training for Large Language Models with Oscillation Suppression and Outlier Control

Yuxiang Chen [1 2]   Yifan Liu [1]   Xiaoming Xu [1]   Pengle Zhang [1]   Michael Beyer [3]   Martin Rapp [3]   Jun Zhu [1]
Jianfei Chen [1]

## Abstract

Large Language Models (LLMs) training is prohibitively expensive, driving interest in low-precision fully-quantized training (FQT). While novel 4-bit formats like `NVFP4` offer substantial efficiency gains, achieving near-lossless training at such low precision remains challenging. We introduce **TetraJet-v2**, an end-to-end 4-bit FQT method that leverages `NVFP4` for activations, weights and gradients in all linear layers. We identify two critical issues hindering low-precision LLM training: weight oscillation and outliers. To address them, we propose: 1) an unbiased double-block quantization method for `NVFP4` linear layers with practically optimal convergence in LLM training, 2) **OsciReset**, the first effective algorithm to suppress LLMs' weight oscillation bottleneck, and 3) **OutControl**, a mix-precision algorithm to retain outlier accuracy. **TetraJet-v2** outperforms prior methods on `FP4` pre-training for LLMs across models up to `370M` parameters trained up to `212B` tokens, reducing the performance gap to `BF16` by an average of 51.3% while enabling $1.67\times$ end-to-end speedup over `FP8`.[1]

## 1. Introduction

Large language models (LLMs) have drastically grown in size in recent years, reaching several hundred billions of parameters (Guo et al., 2025) and are trained on trillions of tokens (Grattafiori et al., 2024). At such scales, the cost of pre-training a single state-of-the-art model can exceed 100 million US dollars (Maslej et al., 2025), which is prohibitively expensive. Leveraging low-precision computations has emerged as a powerful means to reduce the computational resources and memory requirements for training neural networks. This is achieved by performing both the forward and the backward pass with quantized tensors (i.e., activations, weights, and gradients), which can be computed more efficiently on modern hardware, resulting in higher throughput. Training using 16-bit floating-point formats such as `FP16` (Narang et al., 2017) or `BF16` (Kalamkar et al., 2019), as well as 8-bit formats such as `FP8` (Liu et al., 2024; Peng et al., 2023) is increasingly mature and has reached commercial deployment (Liu et al., 2024), and these efforts are enabled in part by modern hardware support, such as `FP8` on NVIDIA Hopper (Choquette, 2023).

Recently, *microscaling* formats for 4-bit representations, such as `MXFP4` (Rouhani et al., 2023) and `NVFP4` (Alvarez et al., 2025), have been introduced into modern hardware, e.g., NVIDIA Blackwell architecture (Tirumala & Wong, 2024). Unlike traditional low-precision formats that apply a single global scaling factor to an entire tensor, microscaling formats assign separate scaling factors to smaller groups of values (e.g., 32 for `MXFP4` and 16 for `NVFP4`). This fine-grained quantization significantly reduces quantization error, especially for tensors containing outlier elements, with a slight increase in control complexity at runtime. Among these formats, `NVFP4` stands out as the most accurate `FP4` format, delivering up to $3\times$ higher compute performance and $2\times$ lower memory usage compared to `MXFP8` (NVIDIA et al., 2025), greatly improving training efficiency.

Prior work on post-training quantization has demonstrated that LLMs can maintain their original task performance even at low-precision (e.g., 4-bit (Ashkboos et al., 2024)). However, achieving *stable end-to-end* 4-bit training has proven more challenging. Early attempts at fully quantized 4-bit training suffered from severe performance degradation (Sun et al., 2020; Chmiel et al., 2023; Xi et al., 2023). More recently, `FP4` pre-training work adopts the modern data format `MXFP4`/`NVFP4`, and combines several techniques trying to mitigate outlier and instability problems in quantization training, such as Hadamard Rotation (Tseng et al., 2025; Castro et al., 2025), and differentiable gradient

---

[1]Dept. of Comp. Sci. and Tech., Institute for AI, BNRist Center, THBI Lab, Tsinghua-Bosch Joint ML Center, Tsinghua University [2]Zhili College, Tsinghua University [3]Bosch AI Research, Renningen, Germany. Correspondence to: Jianfei Chen <jianfeic@tsinghua.edu.cn>.

*Proceedings of the $43^{rd}$ International Conference on Machine Learning*, Seoul, South Korea. PMLR 306, 2026. Copyright 2026 by the author(s).

[1]Code: https://github.com/thu-ml/TetraJet-v2-NVFP4Training

estimation (Wang et al., 2025). In particular, NVIDIA's NVFP4 pre-training recipe (NVIDIA et al., 2025) demonstrates the *practical feasibility* of large-scale FP4 training by leveraging techniques such as 2D weight quantization and Hadamard transforms. However, their approach is not fully 4-bit, as a subset of Transformer blocks is kept in BF16 throughout training to maintain stability. Moreover, the design of their linear layers is not yet optimal for fully FP4 training. As a result, *fully* FP4 training methods remain not comparable to high-precision training in our experiments, and the challenging optimization problems lying in the FP4 training remain unsolved.

In this work, we find that the performance gap between fully low-bit training and high-precision training cannot be explained by quantization error alone, but is also caused by optimization issues introduced by low-precision training. Specifically, we identify two major effects that critically hinder fully FP4 training: 1) *Weight oscillation*, the dominant failure mode in FP4 training, where small updates to high-precision master weights repeatedly flip their quantized representations between adjacent bins. This phenomenon severely disrupts optimization and leads to substantial degradation in downstream performance (Nagel et al., 2022). 2) *Outlier features*, where few channels in activations have massive magnitudes that lead to high dynamic ranges that FP4 cannot represent accurately, which exacerbates optimization difficulty under low precision (Sun et al., 2024).

In this paper, we propose **TetraJet-v2**, a novel low-precision training method for LLMs that explicitly targets the optimization challenges of fully FP4 training. Compared to TetraJet (Chen et al., 2025b), which focuses on MXFP4 training for Vision Transformers, our work targets NVFP4 LLM training and systematically addresses the dominant optimization failures, weight oscillation, and outlier sensitivity. We employ NVFP4 for activations, weights, and gradients of *all* linear layers, enabling an evaluation of *fully* NVFP4 training under modern hardware support. We introduce targeted algorithmic methods that stabilize fully FP4 optimization and substantially narrow the performance gap to high-precision training. We further implement efficient kernels for our methods, observing only minimal overhead.

In summary, our contributions are as follows:

- An **unbiased double-block quantization** method for NVFP4 linear layers with unbiased gradient estimation, and the currently optimal fully-FP4 configuration combined with Random Hadamard Transform (RHT) for the LLM training scenario through extensive ablations.

- **OsciReset**, which is the *first* algorithm to resolve LLMs' weight oscillation problem, a primary source of training instability for low-precision models.

- **OutControl**, which further enhances performance by

controlling outliers in both the forward and backward.

- Extensive evaluation of *fully* NVFP4 pre-training on OLMo-2 (up to 370M params, 212B tokens), narrowing the performance gap to full precision by 51.3%. With *all* proposed methods integrated, our CUDA kernels achieve an end-to-end speedup $1.67\times$ over FP8.

**Conflict of Interest Disclosure**  The authors declare no financial or other substantive conflicts of interest.

## 2. TetraJet-v2's NVFP4 Linear Layer Design

### 2.1. Preliminary

**NVFP4 Format**  Each floating-point (FP) number consists of a sign-bit, exponent-bits, and mantissa-bits. An FP format is written as E$x$M$y$ if it has $x$ exponent-bits, and $y$ mantissa-bits. The most common 4-bit floating-point (FP4) is E2M1, whose values are FP4Values $= \{0, \pm 0.5, \pm 1, \pm 1.5, \pm 2, \pm 3, \pm 4, \pm 6\}$.

To represent a matrix in FP4 precision, we need additional scaling factors to scale the numerical range to $[-6, 6]$. The MXFP4 and NVFP4 formats achieve this by partitioning each matrix into groups of 32 and 16 elements, respectively, and they use an 8-bit scaling factor for each group to scale the elements. Specifically, MXFP4 uses an unsigned E8M0 scaling factor, while NVFP4 utilizes the more precise E4M3 scaling factor. Given the finer-grained group shape and the more precise scaling format of NVFP4, we select NVFP4 as our training format.

**Quantization**  To quantize a high-precision (e.g., BF16) matrix to NVFP4, we need to compute the 8-bit scaling factor $S$ for each group and 4-bit E2M1 value $P_i$ for each group element $X_i$. For any group of high-precision values $\{X_i\}_{i=0}^{15}$, we map each $X_i$ to a FP4 values as follows, where $\text{round}_{\text{FP4}}$ can be deterministic or stochastic.

$$P_i = \text{round}_{\text{FP4}}\left(X_i/S\right), \quad X_i \approx P_i \cdot S$$

We adopt *Deterministic Rounding* (Round-To-Nearest, RTN) in forward to minimize quantization error:

$$\text{round}_{\text{FP4},D}(x) = \arg\min_{q \in \text{FP4Values}}\{|x - q|\}$$

We use *Stochastic Rounding* (Courbariaux et al., 2015) in backward to ensure unbiased estimation of gradients. For any $-6 \leq x \leq 6$ we can find two consecutive FP4 values $q_1, q_2 \in \text{FP4Values}$, such that $q_1 \leq x < q_2$. We round $x$ to either $q_1$ or $q_2$ with probability by generating $\xi \sim \text{Uniform}\left(-\frac{q_2-q_1}{2}, \frac{q_2-q_1}{2}\right)$:

$$\text{round}_{\text{FP4},S}(x) = \begin{cases} q_1, & x + \xi \leq \frac{q_1+q_2}{2} \\ q_2, & \text{otherwise} \end{cases}$$

Stochastic rounding is unbiased: $\mathbb{E}_\xi[\text{round}_{\text{FP4},S}(x)] = x$.

## 2.2. Double-Block Quantization to NVFP4 formats

However, the range of the 8-bit E4M3 scaling factor of NVFP4 is only $[-448, 448]$. This is too small to represent large values, so values $X_i$ should be scaled into $[-448 \times 6, 448 \times 6]$ before quantization with the NVFP4-quantizer. Following the numerical limitation of NVFP4, we add a $1 \times 128$-*outer-block* outside the $1 \times 16$-*inner-block*. For an outer-block $\{X_i\}_{i=0}^{127}$, we need to scale values to $[-448 \times 6, +448 \times 6]$ with a global scale factor $S_{\text{global}}$, and then cast to NVFP4 with group quantization for each inner-block:

$$S_{\text{global}} = \frac{\max_{i=0}^{127} |X_i|}{448 \times 6}, \ S_{\text{block}k} = \frac{\max_{i=16k}^{16k+15} \left| \frac{X_i}{S_{\text{global}}} \right|}{6},$$

$$P_i = \text{round}_{\text{FP4}} \left( \frac{X_i}{S_{\text{global}} \times S_{\text{block}\lfloor i/16 \rfloor}} \right),$$

$$X_i \approx P_i \times S_{\text{global}} \times S_{\text{block}\lfloor i/16 \rfloor}.$$

In this way, we can not only satisfy the numerical needs but also avoid computing the maximum absolute value on a full tensor. This is more hardware-friendly and also more accurate compared to NVIDIA et al. (2025), where a per-tensor second quantization scaling is used. In Sec. D.1, we demonstrate that our finer outer-block scaling yields better performance through experiments.

## 2.3. NVFP4 Linear Layers with Unbiased Gradients

The forward/backward pass of a linear layer with input $\mathbf{X}$ and weight $\mathbf{W}$ can be demonstrated as:

$$\mathbf{Y} = \mathbf{X} \times \mathbf{W}^\top, \quad d\mathbf{X} = d\mathbf{Y} \times \mathbf{W}, \quad d\mathbf{W} = d\mathbf{Y}^\top \times \mathbf{X}$$

To accelerate training, we need to compute all three matrix multiplications (MMs) in linear layers with FP4. To achieve this, our method quantizes the six input matrices of the three MMs to NVFP4, following TetraJet's design for MXFP4 linear layer (Chen et al., 2025b), which can be formulated as:

$$\mathbf{Y} = \widehat{\mathbf{X}} \times \widehat{\mathbf{W}^\top}, \ \widehat{\mathbf{X}} := Q_D^{(1)}(\mathbf{X}), \ \widehat{\mathbf{W}^\top} := Q_D^{(2)}(\mathbf{W}^\top)$$

$$d\mathbf{X} = Q_S^{(3)}(d\mathbf{Y}) \times Q_S^{(4)}\left(\widehat{\mathbf{W}}\right),$$

$$d\mathbf{W} = Q_S^{(5)}\left(d\mathbf{Y}^\top\right) \times Q_S^{(6)}\left(\widehat{\mathbf{X}}\right)$$

where $\mathbf{X} \in \mathbb{R}^{N \times D}, \mathbf{W} \in \mathbb{R}^{C \times D}, \mathbf{Y} \in \mathbb{R}^{N \times C}, \ \widehat{\mathbf{W}} := \widehat{\mathbf{W}^\top}^\top$, and $d\mathbf{X}, d\mathbf{Y}, d\mathbf{W}$ are the input/output/weight gradient matrices respectively referring to $\nabla_{\mathbf{X}} \mathcal{L}, \nabla_{\mathbf{Y}} \mathcal{L}, \nabla_{\mathbf{W}} \mathcal{L}$ ($\mathcal{L}$ is the loss function), and $Q_{D/S}$ represents the deterministic/stochastic rounding. To meet the acceleration need of MMs, the group shape of $Q^{(1)}, Q^{(3)}, Q^{(5)}$ should be $1 \times 16$ and $Q^{(2)}, Q^{(4)}, Q^{(6)}$ should be $16 \times 1$.

Our linear layer design has three major differences compared to NVIDIA's training recipe (NVIDIA et al., 2025):

1. **The alignment of tensors in forward and backward**: In the computation of $d\mathbf{W}$, the correct gradient should be computed upon $\widehat{\mathbf{X}}$ rather than $\mathbf{X}$ to align with the forward. Therefore, we choose to estimate $\widehat{\mathbf{X}}$ while NVIDIA et al. (2025) estimates $\mathbf{X}$ in backward.

2. **The unbiasedness from stochastic rounding**: We adopt stochastic rounding to achieve unbiased estimation in backward, while NVIDIA et al. (2025) chooses deterministic rounding for $\mathbf{X}$ in the backward, which may lead to bias in gradient expectation.

3. **The block shape of weights**: We choose a more accurate $1 \times 16$ shape, while NVIDIA et al. (2025) adopts $16 \times 16$ as weight group shape. Our non-square block shape needs alignment for $\widehat{\mathbf{W}}$, so $\widehat{\mathbf{W}}$ rather than $\mathbf{W}$ is unbiasedly estimated in the computation of $d\mathbf{X}$.

Our design ensures the unbiasedness of gradients estimation according to a similar analysis in TetraJet (Chen et al., 2025b), theoretically ensuring the convergence of SGD (Chen et al., 2020). This results in better performance in the NVFP4 training compared to previous work. We show the superiority of our design in Sec. 5.2.

# 3. OsciReset: Oscillation Suppression Through Resetting Master Weights

## 3.1. Oscillation Phenomenon in LLMs Pre-training

Although the unbiased gradient makes NVFP4 training feasible, the optimization of low-precision weights is not the same as that of high-precision weights. In the final stage of low-precision training, the learning rate (LR) drops to near zero, so the model can quickly descend to a local minimum for convergence. However, we found that when the master weights converge as the LR is approaching zero, there are still drastic changes among the quantized weights. This phenomenon is *weight oscillation*, which widely exists in quantized training.

To demonstrate this phenomenon, we plot the distribution of *latent weight* $w/s$ for linear weight parameters during NVFP4 training, where $s$ is $w$'s quantization scaling factor. According to Fig. 1a, there is a growing number of latent weights lying close to the quantization decision threshold. It means that more and more weights are prone to switching quantization value. This trend explains the drastic changes the quantized model exhibits at the end of training.

For example, $q_1 = 0, q_2 = 0.5$ are two FP4 values. The latent weights near the threshold $\frac{q_1+q_2}{2} = 0.25$ would be sensitive to small perturbations. Assume $w/s \in (0.25, 0.25 + \varepsilon)$, where $\varepsilon$ is a small number. The corresponding quantized weight is $q_2 = 0.5$. Even a tiny gradient step could make $w/s$ jump to $w/s \in (0.25 - \varepsilon, 0.25)$, re-

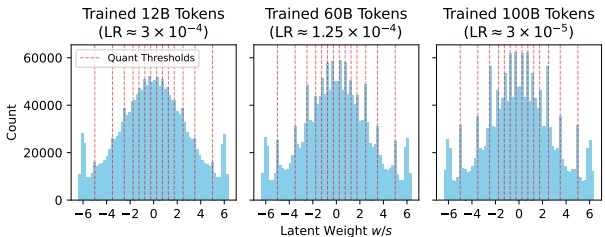

*(a)* Distribution of latent weight $w/s$ in `OLMo2-150M` `blocks.11.att_proj` without oscillation suppression.

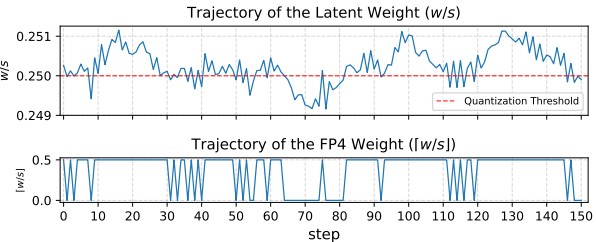

*(b)* Trajectory of an oscillating weight near the end of training.

*Figure 1.* Weight oscillation behavior in `NVFP4` training.

sulting in a giant leap to $q_1 = 0$, and vice versa. This process can frequently repeat for the weights near the threshold. In Fig. 1b, we can see a trajectory of an oscillating weight — the latent value $w/s$ is always near and frequently crossing the threshold $\frac{q_1+q_2}{2} = 0.25$, so the quantized values switch frequently between $q_1 = 0$ and $q_2 = 0.5$.

If we want to reach a low-bit model ultimately, the oscillation phenomenon needs to be addressed, for the oscillating weights cannot converge well to a determined quantized value and will become a potential detriment to the global optimization. Therefore, we aim to identify the specific elements responsible for oscillations during LLM training, and apply additional mechanisms to suppress oscillations, thereby improving the overall optimization performance.

### 3.2. Identifying Oscillating Weights

Following the previous oscillation detection method (Chen et al., 2025b) for Vision Transformers, we identify weight oscillation through tracking the optimization trajectory of each weight element. During our tracking in an interval $t = 0, 1, 2, \ldots, T_0$, we record the optimization distance of master weights and quantized weights, respectively, for each weight element:

$$\text{dist}_M(w) = \sum_{t=1}^{T_0} |w^{(t)} - w^{(t-1)}|,$$

$$\text{dist}_Q(w) = \sum_{t=1}^{T_0} |Q(w^{(t)}) - Q(w^{(t-1)})|$$

---

**Algorithm 1** OscillationSuppress($\theta$, $\text{dist}_M$, $\text{dist}_Q$, $\tau_{\text{osci}}$)

**Input:** $\theta$: model parameters;
   $\tau_{\text{osci}}$: oscillation risk threshold;
   $\text{dist}_M, \text{dist}_Q$: oscillation statistics.
**Output:** Modified parameters $\theta$
**for** $i$-th Element $w_i$ **in** Quantized Parameters of $\theta$ **do**
   Calculate OsciRisk: $\text{OsciRisk}(w_i) \leftarrow \frac{\text{dist}_Q(w_i)}{\text{dist}_M(w_i)}$
   Get Quantization Information:
      $w_{\text{FP4},i} \leftarrow$ Quantized FP4 Value of $w_i$
      $\text{scale}_i \leftarrow$ Scaling Factor of $w_i$
   **if** $\text{OsciRisk}(w_i) \geq \tau_{\text{osci}}$ **then**
      Identify $w_i$ as an oscillating weight;
      Reset $w_i$ to the **bin center**: $w_i \leftarrow w_{\text{FP4},i} \cdot \text{scale}_i$
   **end if**
**end for**
**Return:** $\theta$

---

The risk of oscillation can be defined as $\text{OsciRisk}(w) = \text{dist}_Q(w)/\text{dist}_M(w)$. A larger $\text{OsciRisk}(w)$ indicates that the weight element $w$ oscillates because its quantized weight is switching frequently ($\text{dist}_Q(w)$ is relatively large) but its master weight is moving in small steps ($\text{dist}_M(w)$ is relatively small).

For larger-scale, we provide a novel *memory-efficient* implementation by tracking only the top-5% weights closest to thresholds (adds $\sim 0.6$ Bytes/param). We store an index along with four subset-tracking states (previous master/quantized weight and accumulated distance). This maintains efficacy because weights far from thresholds rarely oscillate. In distributed training, the states are naturally sharded across GPUs, further reducing the per-device memory footprint. We represent the detailed calculation process in Alg. 2 & Alg. 3 and more analysis in Appendix C.

### 3.3. Reducing Oscillation by Resetting Master Weights

After identifying those special elements prone to oscillation, we would stabilize them for improving convergence. This is a challenging task, because suppressing oscillation would likely harm global optimization, which is why previous methods for Vision Transformers cannot generalize well to LLM (as shown in Sec. 5.2). We need a method to effectively stabilize the oscillation, and meanwhile, to maintain a good optimization for all the parameters.

Our novel stabilization method solves this problem by simply resetting the oscillating weight to the center of the quantization bin where it is lying, which is Alg. 1.

We adopt this algorithm when the learning rate decays to a relatively low value and oscillation problems emerge. The algorithm has the following effects:

- The quantized weights are not affected right after the

---

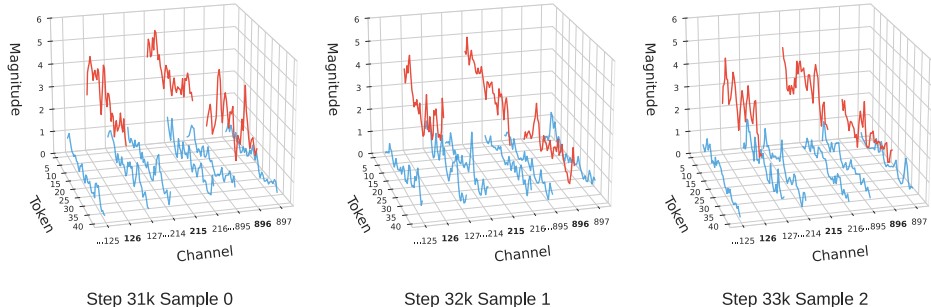

*Figure 2.* Activation magnitudes of MLP input at layer 10 for different `GSM8K` samples across `OLMo2-370M` training checkpoints at different steps. Outliers consistently appear in specific channels.

master weights are reset, ensuring *no performance degradation* from the resetting.

- If there is not such resetting, oscillating elements would keep fluctuating between two fixed bins, so *resetting* can provide more chances to be optimized to other values. This is better than *freezing*, which results in these parameters never being updated in the future.

In this way, the oscillating weights can be *reset* to follow a new optimization trajectory less prone to oscillation, reaching an improved model performance. Combining the suppression algorithm, the refined trainer **OsciReset** can be implemented as Alg. 4 in Appendix C.

Prior methods on weight oscillation mainly target Vision Transformers, require careful hyper-parameter tuning, and may only apply to fine-tuning from full-precision checkpoints (Nagel et al., 2022; Liu et al., 2023; Chen et al., 2025b). Our method, in contrast, is simple and directly applicable to LLM pre-training from scratch with minimal tuning. We further show in the Appendix C.2 that it introduces negligible latency and memory overhead.

## 4. OutControl: Outlier Control for Activation and Gradients

Large *outliers* are widely present in the activations and gradients of LLMs, but are harmful for quantization. Due to the presence of outliers which results in excessively large scaling factors, small numbers in the quantization group would be scaled down near zero, and cannot be represented accurately through low-precision formats like `FP4`.

### 4.1. Random Hadamard Transformation for Quantized Backpropagation

**Hadamard Matrix** A common way to reduce outliers is performing orthogonal rotation on the tensors with the Hadamard matrix (Halko et al., 2011). Hadamard matrix

$H_d(d = 2^k, k \in \mathbb{Z}_+)$ is an orthogonal matrix defined by

$$H_2 = \frac{1}{\sqrt{2}} \begin{bmatrix} 1 & 1 \\ 1 & -1 \end{bmatrix}, \quad H_{2d} = \frac{1}{\sqrt{2}} \begin{bmatrix} H_d & H_d \\ H_d & -H_d \end{bmatrix}$$

Intuitively, Hadamard rotation makes the distribution of a tensor more even, mitigating large outliers. In practice, we use a block Hadamard $\mathbf{H}_n = \text{diag}\{H_d, H_d, \cdots, H_d\} \in \mathbb{R}^{n \times n}$ ($d \in \{16, 32, \cdots\}, d|n$) instead of a full Hadamard matrix to avoid heavy computation (Xi et al., 2023).

**Random Hadamard Transformation** To control the quantization variance in the backward pass, we follow Tseng et al. (2025) and apply a Random Hadamard Transformation (RHT). The RHT performs a random sign flipping before Hadamard Transformation: $\mathbf{A}' \leftarrow \mathbf{AS}_n\mathbf{H}_n$, where $\mathbf{S}_n \in \text{diag}\{\pm 1\}^n$ is randomly generated. Applying RHT simultaneously to both operands of a matrix multiplication (`MM`) results in an *equivalent* result since $\mathbf{H}_n$ and $\mathbf{S}_n$ are orthogonal: $\mathbf{AB}^T = (\mathbf{AS}_n\mathbf{H}_n)(\mathbf{BS}_n\mathbf{H}_n)^T$. Therefore, we can alleviate matrices' outliers through RHT before low-precision `MM` as: $\mathbf{AB}^T \approx Q(\mathbf{AS}_n\mathbf{H}_n)Q(\mathbf{BS}_n\mathbf{H}_n)^T$.

**Our Design: Apply RHT to** $d\mathbf{X}$ **and** $d\mathbf{W}$ **Computation** As described in Sec. 2.3, there are one `MM` in the forward and two `MM`s in the backward of a linear layer. Practically, we apply RHT for the two `MM`s in backward, the computation of $d\mathbf{X}$ and $d\mathbf{W}$. This design is based on our empirical findings (Sec. 5.2). We do not adopt the Hadamard transformation in the forward pass like Castro et al. (2025) because we observe that Hadamard transformation to the forward pass is harmful for the optimization. We believe that the additional error from RHT exceeds the outlier reduction effect in the forward. Moreover, while NVIDIA et al. (2025) suggests RHT brings benefits only for $d\mathbf{W}$, our experiments yield a different insight that RHT would improve both computation of $d\mathbf{X}$ and $d\mathbf{W}$. Therefore, we apply RHT to all `MM`s in backward passes in `NVFP4` linear layers.

*Table 1.* Training and validation perplexity of `OLMo2` pre-training with different `FP4` training methods. The `70M`/`150M`/`370M` models are trained from scratch with `52B`/`107B`/`212B` tokens.

| METHODS \ OLMo2-SIZE | TRAIN PPL | | | VALIDATION PPL | | |
|---|---|---|---|---|---|---|
| | 70M | 150M | 370M | 70M | 150M | 370M |
| BF16 | 35.95 | 26.38 | 18.70 | 45.27 | 33.49 | 23.70 |
| QUARTET | 40.77 | 29.25 | 20.76 | 51.23 | 36.89 | 26.16 |
| NVIDIA | 40.50 | 29.18 | 20.75 | 50.94 | 36.73 | 26.20 |
| TETRAJET-V2-BASE (OURS) | 39.26 | 28.39 | 20.23 | 49.33 | 35.88 | 25.50 |
| TETRAJET-V2-FULL (OURS) | **38.08** | **27.58** | **19.89** | **47.75** | **34.95** | **25.11** |

## 4.2. Precision Retaining for Activation Outliers

Besides adopting RHT in the backward pass to improve the gradient calculation, we observe that RHT performs poorly in the forward pass compared to without RHT as described above. To overcome this limitation and further boost performance, we introduce a method based on retaining the precision of certain outliers in activations.

We note that the most obvious outlier pattern in activations is *structural*. This means that a small number of activation channels *consistently* exhibit much larger variance (higher norms) than others across different inputs and training steps, as shown in Fig. 2, echoing observations in Xiao et al. (2023). Therefore, we can leverage this fixed structural pattern to statically select these persistent outlier channels and retain them in higher precision (e.g., FP8, BF16). Moreover, for backward pass, we further combine it with the RHT technique to further control outliers beyond selected channels, ensuring additional gradient accuracy.

This method is inspired by `LLM.int8()` (Dettmers et al., 2022), a PTQ technique that identifies and retains activation outlier channels during inference. However, unlike `LLM.int8()`, our method applies to the fully low-precision training, simultaneously improving the accuracy of forward and backward passes.

In detail, we identify the outlier index set $\mathcal{A}$ by accumulating the activation $\ell_2$-norms over a brief calibration window (e.g., 50 steps) triggered early in training (e.g., at 1% progress). Based on the accumulated statistics, we select the top $p\%$ channels as $\mathcal{A}$ and use $\bar{\mathcal{A}}$ for the complement indices. Typically, we choose $p = 10\%$ in FP8 formats. The forward and backward passes are then computed as follows:

$$\mathbf{Y} = Q_D^{(1)}(\mathbf{X}_{:,\bar{\mathcal{A}}}) \times Q_D^{(2)}(\mathbf{W}_{\bar{\mathcal{A}}}^\top) + \underline{\mathbf{X}_{:,\mathcal{A}} \times \mathbf{W}_{\mathcal{A}}^\top}$$

$$\mathrm{d}\mathbf{X} = Q_S^{(3)}\left(\mathrm{d}\mathbf{Y}\mathbf{S}_C\mathbf{H}_C\right) \quad \times Q_S^{(4)}\left(\mathbf{H}_C^\top\mathbf{S}_C^\top\widehat{\mathbf{W}}\right),$$

$$\mathrm{d}\mathbf{W}_{:,\bar{\mathcal{A}}} = Q_S^{(5)}\left(\mathrm{d}\mathbf{Y}^\top\mathbf{S}_N\mathbf{H}_N\right) \times Q_S^{(6)}\left(\mathbf{H}_N^\top\mathbf{S}_N^\top\widehat{\mathbf{X}}_{:,\bar{\mathcal{A}}}\right),$$

$$\mathrm{d}\mathbf{W}_{:,\mathcal{A}} = \underline{\mathrm{d}\mathbf{Y}^\top \times \widehat{\mathbf{X}}_{:,\mathcal{A}}}$$

Here $\widehat{\mathbf{X}}_{:,\bar{\mathcal{A}}} := Q_D^{(1)}(\mathbf{X}_{:,\bar{\mathcal{A}}})$, $\widehat{\mathbf{X}}_{:,\mathcal{A}} := \mathbf{X}_{:,\mathcal{A}}$ represents the

aligned activation used in backward pass, and $\widehat{\mathbf{W}}_{\bar{\mathcal{A}}} := Q_D^{(2)}(\mathbf{W}_{\bar{\mathcal{A}}}^\top)^\top, \widehat{\mathbf{W}}_{\mathcal{A}} := \mathbf{W}_{\mathcal{A}}$ represents the aligned weight. Here the precision of $\mathbf{W}_{\mathcal{A}}$ can be either high precision (e.g.,FP8,BF16) or FP4 (to produce a fully FP4-weighted model). The high-precision multiplications are in red color.

As mentioned above, our outlier selection for $\mathbf{X}$ is offline, since outlier patterns are consistent across inputs and steps. This consistency enables us to statically select outlier channels at the starting stage of training, removing the need for dynamic adjustments for each input batch or training step. Besides, this consistency stabilizes the model structure by fixing the selected channels and aligns the model for inference with training. Compared to previous outlier selection methods like Wang et al. (2025), which require input-specific dynamic adjustments, our static selection method incurs minimal computational overhead in maintaining outlier structure.

## 5. Experiments

### 5.1. Pre-Training LLMs with NVFP4

**Model and Data Settings** Based on the official open-source code base of OLMo[2], we pretrained `OLMo2` models (OLMo et al., 2024) with 70M, 150M, and 370M parameters[3], from scratch for 52B, 107B, and 212B tokens respectively, on the open-sourced `OLMo-2-Mix-1124` dataset[4]. We adopt the AdamW optimizer (Loshchilov & Hutter, 2019) with a cosine-decay learning-rate scheduler with warm-up for all pre-training. The details of the models and training recipes are listed in Appendix B.1.

**NVFP4 Algorithm Settings** Following prior work (Xi et al., 2023), we focus on evaluating the quantization algorithm for the *linear* layers, and quantize activation, weight, and gradients to achieve *Fully Quantized Training* (FQT). For a fair comparison, we enforce a strict fully 4-bit setting where *all* linear layers in Transformer (Vaswani et al., 2017)

---

[2]https://github.com/allenai/OLMo
[3]The number of parameters here is excluding embeddings.
[4]https://huggingface.co/datasets/allenai/olmo-mix-1124

*Table 2.* Performance on downstream tasks of `OLMo-2 370M` trained for `212B` tokens with different `FP4` training methods.

| Methods | ARC-e | ARC-c | BoolQ | CQA | Hella. | MMLU | OBQA | PIQA | SIQA | Wino. | Avg.↑ | Wiki. | Pile |
|---|---|---|---|---|---|---|---|---|---|---|---|---|---|
| | acc↑ | acc_n↑ | acc↑ | acc_n↑ | acc_n↑ | acc_n↑ | acc_n↑ | acc_n↑ | acc_n↑ | acc↑ | | PPL↓ | PPL↓ |
| BF16 | 55.96 | 28.43 | 54.49 | 36.60 | 43.30 | 25.85 | 33.40 | 68.66 | 42.78 | 51.86 | 44.13 | 16.86 | 12.21 |
| Quartet | 53.04 | 27.43 | 56.54 | 34.23 | 39.89 | 24.12 | **32.20** | 67.25 | 42.32 | 50.91 | 42.79 | 19.03 | 13.53 |
| NVIDIA | 52.11 | 26.76 | 56.48 | 34.72 | 39.81 | 25.10 | 31.20 | 65.99 | **44.11** | **51.46** | 42.77 | 19.06 | 13.52 |
| TetraJet-v2-base (ours) | 54.56 | **27.75** | **59.32** | 35.38 | 40.53 | 23.75 | 31.00 | **67.57** | 42.89 | 51.32 | 43.41 | 18.52 | 13.16 |
| TetraJet-v2-full (ours) | **54.74** | 24.75 | 59.23 | **37.18** | **41.11** | **26.08** | 31.60 | 66.59 | 43.65 | 51.06 | **43.60** | **18.06** | **12.81** |

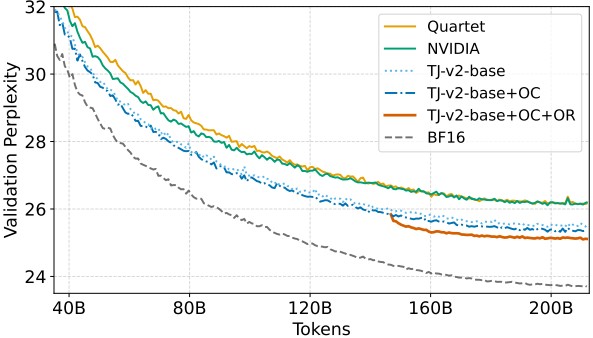

*Figure 3.* Validation loss curve of `OLMo2-370M` with `212B` tokens for comparing different methods. (`TJ-v2`: training method TetraJet-v2; `OC`: OutControl; `OR`: OsciReset)

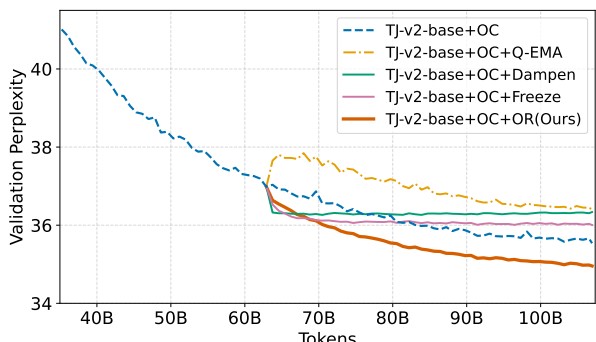

*Figure 4.* Validation loss curve of `OLMo2-150M` with `107B` tokens for different oscillation suppressing techniques. We set $T_{\text{start}} \approx 65B$ for all methods to begin suppressing oscillation.

blocks are quantized. We apply this standard even to the NVIDIA baseline (NVIDIA et al., 2025) (originally mixed-precision) to evaluate robustness under full quantization.

We compared our method with recent state-of-the-art approaches: `MXFP4` training method Quartet (Castro et al., 2025), and NVIDIA's `NVFP4` training recipe (NVIDIA et al., 2025). For our methods, we used two training recipes for a clearer comparison: (1) **TetraJet-v2-base**: combines our `NVFP4` layer with RHT in backward; (2) **TetraJet-v2-full**: adds oscillation suppression (*OsciReset*) and outlier controlling (*OutControl*) over TetraJet-v2-base.

**Results** We present the training perplexity (PPL) (averaged on the last 50 steps) and the validation PPL on the `C4` dataset (Dodge et al., 2021) in Tab. 1. We observe that TetraJet-v2-full reaches the lowest PPL, further closing the gap to high-precision training. For the downstream evaluation, we reported the results in Tab. 2 on the following benchmarks: ARC (Clark et al., 2018), BoolQ (Clark et al., 2019), Commonsense QA (CQA) (Talmor et al., 2019), Hellaswag (Zellers et al., 2019), MMLU (Hendrycks et al., 2020), Open Book QA (OBQA) (Mihaylov et al., 2018), PIQA (Bisk et al., 2020), SocialIQA (SIQA) (Sap et al., 2019), and Winogrande (Sakaguchi et al., 2021). We also report the validation PPL on Wikitext-103 (Merity et al., 2016) and Pile (Gao et al., 2020). TetraJet-v2-full reaches

the highest average performance among all FP4 methods.

### 5.2. Ablation Studies

#### 5.2.1. BASIC NVFP4 LINEAR LAYER SETTING

We conducted detailed experiments on how to select the quantization block size, whether to use aligned activation $\hat{\mathbf{X}}$ and stochastic rounding to yield unbiased gradients, and the effect of the Hadamard Transform on each MM. We show the detailed results in Appendix D.1. As a result, our `NVFP4` linear surpasses NVIDIA's `NVFP4` design (NVIDIA et al., 2025) with a more refined scaling method and unbiased gradient estimation.

#### 5.2.2. IMPROVEMENT OF OSCILLATION SUPPRESSION AND OUTLIER CONTROL

**The Combination of Methods** We show in the validation loss curve in Fig. 3 that our methods OsciReset and OutControl can be finely combined. From the figure, we see that **OutControl** can improve accuracy throughout the training process, and the oscillation suppression algorithm **OsciReset** could effectively alleviate oscillation at $T_{\text{start}} \approx 150B$ tokens to bring additional enhancement.

**Ablations on Oscillation Suppression** The difficulty of suppressing oscillation in LLMs lies in simultaneously con-

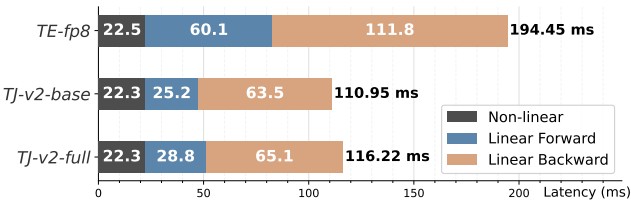

*Figure 5.* End-to-end latency of a Transformer Layer. We test on an RTX 5090 with `MicroBatchSize=4`, `SeqLen=1024`, and `Hidden=16384`. (`TE`: TransformerEngine, `TJ`: TetraJet, `Non-linear`: other components like norm and self-attention.)

*Table 3.* Loss decomposition for quantizers on `OLMo2-370M` for 52B tokens.

| Settings | PPL | ΔPPL |
|---|---|---|
| BF16 | 24.06 | 0 |
| Only `fwd.X` | 24.50 | +0.44 |
| Only `fwd.W` | 24.28 | +0.22 |
| Only `bwd.dX` | 24.13 | +0.07 |
| Only `bwd.dW` | 24.05 | −0.01 |
| Only `bwd.dX+dW` | 24.27 | +0.21 |
| Quant All | 24.89 | +0.83 |

*Table 4.* Loss decomposition for modules on `OLMo2-370M` for 52B tokens.

| Settings | PPL | ΔPPL |
|---|---|---|
| Quant All | 24.89 | 0 |
| w/o `qkv` | 24.74 | −0.15 |
| w/o `attn.out` | 24.73 | −0.16 |
| w/o `mlp.ffn1` | 24.60 | −0.29 |
| w/o `mlp.ffn2` | 24.54 | −0.35 |
| All `BF16` | 24.06 | −0.83 |

trolling the oscillating weights and maintaining the effect of global optimization. We tested several prior methods that work on Vision Transformers with tuned hyperparameters: (1) "Q-EMA" (Chen et al., 2025b) that adopts EMA to smooth weight quantization; (2) "Freeze" (Nagel et al., 2022) that tracks weight oscillation frequency and freezes oscillating weights to the running average; (3) "Dampen" (Nagel et al., 2022) that adds $\lambda\|\mathbf{W} - Q(\mathbf{W})\|^2$ to the loss to encourage weights to move away from thresholds. In Fig. 4, they all present severe detriment to the final performance, and only our method OsciReset consistently maintains a substantial improvement to the optimization of LLMs. In Appendix D.2, we further prove our effectiveness by measuring the decrease in the ratio of oscillating weights.

**Ablations on Outlier Selection** Finally, we validate the effectiveness of the outlier selection method OutControl. In Appendix D.2, we show our static choice of the channels is effective and improves both the forward and backward.

### 5.3. Efficiency Results

We implement CUDA kernels for TetraJet-v2 and evaluate the performance on a Transformer layer on RTX5090. As shown in Fig. 5, our method accelerates linear layers by $1.94\times$ (TetraJet-v2-base) and $1.83\times$ (TetraJet-v2-full) in both forward and backward passes, compared to TransformerEngine `FP8` (NVIDIA, 2024). This translates into an end-to-end speedup of $1.75\times$ and $1.67\times$, respectively. Detailed efficiency results are provided in Appendix G.

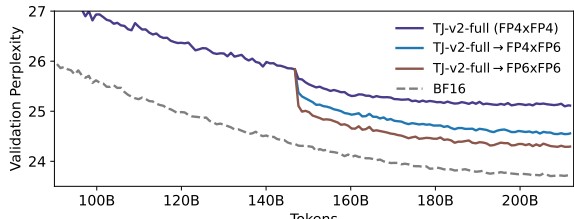

*Figure 6.* Validation Perplexity for the precision switch on `OLMo2-370M`. (We start suppressing oscillation at $T_{\text{start}} \approx$ 150B, so the curve drops for `TJ-v2-full`, `TJ-v2`: TetraJet-v2)

### 5.4. Further Studies and Discussions

**Loss Decomposition** After developing our accurate training method TetraJet-v2-full, with weight oscillation reduction and outlier controlling, we would like to know which quantizer harms the most to the final accuracy, that is, which component of the computation in the forward/backward pass of the linear layer is most sensitive to the quantization. Also, we want to discover which component in the Transformer block is the most sensitive.

In Tab. 3, we find that in the forward pass, the activation would take the most responsibility for the accuracy loss after we adopt weight oscillation. When we only quantize one of the computations of $d\mathbf{X}$ or $d\mathbf{W}$ in backpropagation, it seems lossless. However, if we combine both quantization for gradients, the error would accumulate, leading to a larger accuracy loss. Finally, we conclude that the backward pass is similarly important as the weight quantization, and the activation quantization would be the most crucial bottleneck.

In Tab. 4, linear modules in MLP yields higher sensitivity than those in Attentions. And `mlp.ffn2` is more sensitive to quantization, which may be attributed to the larger outliers brought by SwiGLU (Zhang et al., 2025). We hope the conclusions could inspire future works on low-precision training on how to further improve the performance. We left the decomposition experiments on varying model/data size to future work.

**Precision Switching in Final Stage** Since fully `FP4 × FP4` training remains challenging to be lossless, we explore the accuracy benefits of increasing bit-width (e.g., `FP6 × FP4`, `FP6 × FP6`), despite their limited mainstream hardware support. Here, we simulate a fully `FP6 × FP4` linear layer as follows:

$$\mathbf{Y} = \widehat{\mathbf{X}} \times \widehat{\mathbf{W}^\top}, \ \widehat{\mathbf{X}} := Q_D^{\mathbf{FP6}}(\mathbf{X}), \ \widehat{\mathbf{W}^\top} := Q_D^{\mathbf{FP4}}(\mathbf{W}^\top)$$

$$d\mathbf{X} = Q_S^{\mathbf{FP6}}(d\mathbf{Y}) \times Q_S^{\mathbf{FP4}}\left(\widehat{\mathbf{W}}\right),$$

$$d\mathbf{W} = Q_S^{\mathbf{FP4}}\left(d\mathbf{Y}^\top\right) \times Q_S^{\mathbf{FP6}}\left(\widehat{\mathbf{X}}\right)$$

In Fig. 6, we find that a substantial improvement can be achieved if we just switch to FP6 × FP4 in the last 50B tokens of OLMo2-370M training; also, switching to FP6 × FP6 could bring considerable improvement. We suggest that the FP6 × FP4 training format, though not yet the focus of current mainstream hardware and algorithms, can offer substantial improvement.

## 6. Discussion, Conclusion and Limitations

### 6.1. Discussion

**Major Difference to TetraJet (Chen et al., 2025b) for Vision Transformers.** Firstly, we propose a novel solution to weight oscillation in LLMs (OsciReset). This method needs little tuning and is effective for LLM pre-training while previous methods fail to transfer to LLMs, and it addresses a unique optimization challenge in low-precision training that is completely orthogonal to outlier control. Additionally, we successfully identify the optimal fully-FP4 configuration combined with RHT for the LLM training scenario through rigorous derivation and ablations, whilst TetraJet is only validated on Vision Transformers (ViTs).

Moreover, our evaluation for LLM training on a massive data scale ($500 \sim 600$ token-per-parameter ratio) makes optimization significantly harder and provides a reliable benchmark. In this challenging setting, correctly applying these insights, such as fixing activation misalignment to ensure unbiasedness, and extending outlier retention to both forward and backward passes, is critical for *long-term* convergence when given a large amount of data.

**Why Oscillation-Mitigation Methods from ViTs Do Not Transfer Well to LLMs?** (1) Differences in data/optimization behavior. Image samples only provide a single class label, resulting in a sparse supervision density. A typical ViT-Base trained on ImageNet for 300 epochs only reaches a Target-TPP (Tokens-Per-Parameter) of $\sim4.5$. Moreover, image-classification models can fit even random labels (Zhang et al., 2016), suggesting optimization behavior that differs substantially from long-horizon, token-level LLM training ($> 500$ TPP). Therefore, oscillation-mitigation methods designed for ViTs may no longer be effective for LLM stability. (2) Previous ViT methods have inherent flaws for long-term training. Previous methods harm global optimization or block weights from being optimized further. As shown in paper Fig. 4, their losses would rebound severely after only $\sim5$B tokens.

**The Design of Double-Block Scaling in NVFP4 Quantization.** The limited numerical range of the NVFP4's E4M3 scaling factor ($[-448, 448]$) raises the need of a second scaling factor for tensor quantization. The design in this work is to adopt a $1 \times 128$-sized outer block with an FP32 scaling factor, and we emphasize that it is an implementation detail rather than a main contribution of our work.

The rationale behind our design is that: larger per-tensor scaling needs a global reduction to get the full-tensor absmax, while outer-block (128) scaling can fuse local max into the quantization kernel (avoiding a full extra scan). For larger-LLM future work, it is possible to explore larger granularity (e.g., per-row) to balance scale-overhead and runtime efficiency.

**The Effectiveness of Hadamard Transform (HT).** Egiazarian et al. (2025); Chen et al. (2025a) find that the rotation may hurt NVFP4 quantization and can reduce NVFP4 quality (e.g., lower quantization signal-to-noise ratio), which is consistent with our forward-pass degradation result. Our work analyzes backward quantization separately and finds that, despite forward degradation, Random HT (RHT) improves both dW and dX computation with unbiased gradients, and this finding differs from that from NVIDIA et al. (2025) after we adopt a strictly unbiased backward pass. However, the effectiveness of HT/RHT is still under exploration in other LLM architectures (e.g., MoE) or larger-scale training scenarios.

### 6.2. Conclusion

In this work, we propose **TetraJet-v2**, an end-to-end fully-quantized NVFP4 training recipe for LLMs. Our experiments demonstrate that our unbiased double-block quantization, weight oscillation reduction, and outlier precision control methods enable stable and accurate 4-bit training by solving oscillation and outlier bottlenecks, and consistently outperform previous methods, offering significant advantages for bridging the performance gap to full-precision training while maintaining practical speedup.

### 6.3. Limitations

Due to limited compute resources, our experiments focus on OLMo-2 models with sizes up to 370M parameters and data scales up to 212B tokens. We hope that future work will enable more efficient training of larger-scale LLMs / larger amount of data with efficient algorithms, and explore on more LLM architectures.

## Acknowledgement

This work was supported by the Fundamental and Interdisciplinary Disciplines Breakthrough Plan of the Ministry of Education of China (No. JYB2025XDXM101); the NSFC Projects (Nos. 62595773, 62376131, 62550004, 92270001). J.Z is also supported by the XPlorer Prize.

## Impact Statement

This work improves the efficiency of low-precision LLM training, which may help reduce the hardware cost and energy consumption of large-scale pre-training, thereby making foundation model research more accessible and lowering its carbon footprint. At the same time, improving the efficiency of model training may also lower the barrier to developing and deploying powerful language models, which could facilitate harmful uses such as automated misinformation, spam generation, or misuse at scale. We believe these risks should be mitigated through responsible release practices, usage policies, and continued research on model safety and governance.

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

# A. Related Works

**Low-Precision Training**   Prior work on neural network quantization focused mainly on optimizations that improve inference efficiency of models after training.  To this end, post-training quantization (PTQ) and quantization-aware training (QAT) methods only quantize the forward pass. In contrast, fully quantized training (FQT) aims at improving the computational efficiency during training by quantizing activations, weights, and gradients in the forward and backward pass. The limited dynamic range of 4-bit data types leads to increasing errors and thus numerical instabilities such that multiple methods are leveraged simultaneously to reduce the loss gap between full precision training and FQT. Low-precision training recipes typically use a combination of fine-grained quantization with microscaling data formats, stochastic rounding for unbiased gradient estimates, and outlier compensation strategies such as random Hadamard transforms or clamping to reduce their effect on quantization (Rouhani et al., 2023; NVIDIA et al., 2025; Tseng et al., 2025; Castro et al., 2025; Wang et al., 2025). More recently, some works explore structure-aware and scale-aware approaches, including preserving low-rank components (Cao et al., 2025) and refining quantization scale design for NVFP4 (Cook et al., 2026). While these approaches improve representational accuracy, they do not address optimization challenges that are unique to fully low-bit training. In contrast, our work focuses on mitigating optimization issues such as weight oscillation, which are largely orthogonal to the methods proposed in prior work.

**Weight Oscillation Problem**   Prior work on weight oscillation focuses on convolutional and Transformer-based neural networks in the vision domain and mainly targets QAT, i.e., when fine-tuning a full-precision checkpoint (Nagel et al., 2022; Liu et al., 2023; Chen et al., 2025b). In Nagel et al. (2022) two methods are introduced that address the oscillation during QAT. First, a regularization term that aims at dampening oscillating weights, effectively encouraging them to be closer to the center of a quantization bin. Second, oscillating weights are frozen during QAT. For pre-training this approach is not ideal as these weights will no longer be updated, which limits their contribution toward optimizing the training loss. Recent work on training vision Transformers with MXFP4 introduced the two methods Q-EMA and Q-Ramping, which aim at limiting the magnitude and frequency of updates for oscillating weights (Chen et al., 2025b). Using an exponential moving average (EMA) for weight update steps has been shown to reduce oscillation of weights and the impact thereof on the model's task performance. Further, by adapting the update frequency and using a higher gradient accumulation step, Q-Ramping can effectively reduce the oscillation problem. In this work we expand upon Chen et al. (2025b) and introduce a novel method that addresses weight oscillation during low-precision (pre-)training of LLMs that requires little tuning of hyperparameters.

**Outlier Control**   Several works tried to mitigate outliers with architectural changes, such as gated attention (Qiu et al., 2025) or attention sinks (Xiao et al., 2024). However, these methods cannot fully mitigate outliers, and their quantization remains an open challenge to be addressed at the algorithm level.  The introduction of highly fine-grained block-wise quantization addresses outlier features to some extent, since large values only affect the quantization of a smaller subset of values. In Dettmers et al. (2022), matrix multiplications with outliers features are handled separately in the forward pass. Products involving outliers are computed with higher precision (e.g., FP16), products for remaining values are performed with quantized low-precision 8-bit integer multiplication kernels. Several recent works use random or learned rotations to transform features into a more favorable space that can be quantized easily (Ashkboos et al., 2024; Liu et al., 2025). However, with the transition from PTQ to FQT, the challenge of accurately quantizing outliers has shifted from post-training deployment-related optimizations to pre-training. Recent work on NVFP4 pre-training of LLMs integrates prior methods that have previously mainly been used for PTQ optimization, such as Hadamard transforms for end-to-end low-precision pre-training (NVIDIA et al., 2025; Tseng et al., 2025; Castro et al., 2025).

# B. Details for Reproducing Pre-Training Results of TetraJet-v2

## B.1. Hyperparameters for Model

*Table 5.* Model Hyperparameters in Sec. 5.1: Pre-Training LLMs with `NVFP4`.

| Model | `OLMo2-70M` | `OLMo2-150M` | `OLMo2-370M` |
|---|---|---|---|
| # Parameters (non-embedding) | 72,368,640 | 147,886,848 | 371,262,464 |
| # Parameters (all) | 123,748,864 | 224,957,184 | 474,022,912 |
| Hidden Size | 512 | 768 | 1024 |
| # Heads | 8 | 12 | 16 |
| # Layers | 8 | 12 | 16 |
| # MLP Hidden Size | 2048 | 3072 | 8192 |
| Vocab Size | | 100278 | |
| Sequence Length (N) | | 4096 | |
| Batch Size (BS) | | 1024 | |
| Optimizer | | AdamW | |
| Learning Rate Scheduler | | Cosine Decay with Warm-up | |
| Weight Decay | | 0.1 | |
| Max Learning Rate (LR) | $4 \times 10^{-4}$ | $3 \times 10^{-4}$ | $3 \times 10^{-4}$ |
| # Trained Steps | 12500 | 25500 | 50500 |
| # Warm-up Tokens | 1 B | 8 B | 8 B |
| # Trained Tokens (=Steps×BS×N) | 52 B | 107 B | 212 B |

## B.2. Hyperparameters for NVFP4 Training Techniques

*Table 6.* Hyperparameters for `NVFP4` Training Techniques.

| Technique | Hyperparameters | `OLMo2-70M` | `OLMo2-150M` | `OLMo2-370M` |
|---|---|---|---|---|
| RHT | Hadamard Block Size | | 16 | |
| OsciReset | $T_{\max}$: total number of training steps | 12500 | 25500 | 50500 |
| | $T_{\mathrm{start}}$: suppression start step | 8000 | 15000 | 35000 |
| | $T_{\mathrm{period}}$: suppression period length | | 200 | |
| | $T_{\mathrm{accu}}$: accumulated steps | | 50 | |
| | $\tau_{\mathrm{osci}}$: oscillation risk threshold | | 8 | |
| OutControl | Outlier Compute Type | | FP8 | |
| | Outlier Selected Ratio | | 10% | |

# C. Detailed Implementation and Analysis of Oscillation Suppression

## C.1. Detailed Implementations of OsciReset Algorithms in Sec. 3

---

**Algorithm 2** UpdateOscillationStats($\theta, t_0$)

---

**Input:** $\theta$: model parameters; $t_0$: step in a period.
**Output:** $\mathbf{D}_M, \mathbf{D}_Q$: updated oscillation statistics.
**Data:** $\mathbf{D}_M, \mathbf{D}_Q$: accumulated statistics; $\mathbf{W}'$: weight snapshots.
**if** $t_0 = 0$ **then**
    **for** $k$-th quantized weight matrix $\mathbf{W}_k$ **in** $\theta$ **do**
        Initialize: $\mathbf{D}_{M,k} \leftarrow \mathbf{0}, \mathbf{D}_{Q,k} \leftarrow \mathbf{0}$
        Build record: $\mathbf{W}'_k \leftarrow \mathbf{W}_k$
    **end for**
**else**
    **for** $k$-th quantized weight matrix $\mathbf{W}_k$ **in** $\theta$ **do**
        Compute $Q(\mathbf{W}_k)$ via FP4 quant-dequant
        Update statistics:
$$\mathbf{D}_{M,k} \leftarrow \mathbf{D}_{M,k} + |\mathbf{W}_k - \mathbf{W}'_k|$$
$$\mathbf{D}_{Q,k} \leftarrow \mathbf{D}_{Q,k} + |Q(\mathbf{W}_k) - Q(\mathbf{W}'_k)|$$
        Update record: $\mathbf{W}'_k \leftarrow \mathbf{W}_k$
    **end for**
**end if**
**Return:** $\mathbf{D}_M, \mathbf{D}_Q$
**Note:** $\mathbf{D}_{M/Q}$ are the tensor-wise aggregation of element-wise statistics $\text{dist}_{M/Q}(w_i)$ over all elements $w_i \in \mathbf{W}$.

---

---

**Algorithm 3** UpdateOscillationStats_**MemoryEfficient**($\theta, t_0, p_{\text{sample}}$)

---

**Input:** $\theta$: model parameters; $t_0$: step in a period; $p_{\text{sample}}$: sampling ratio (e.g., $5 \sim 10\%$).
**Output:** $\text{idx}, \widetilde{\mathbf{D}}_M, \widetilde{\mathbf{D}}_Q$: sampled sensitive indices and accumulated distances on $\text{idx}$.
**Data:** $\text{idx}, \widetilde{\mathbf{D}}_M, \widetilde{\mathbf{D}}_Q$; $\widetilde{\mathbf{W}'}_M, \widetilde{\mathbf{W}'}_Q$: master / quantized snapshots on $\text{idx}$.

**Note:** $\text{binwidth}(\cdot)$ returns the quantization bin width corresponding to an FP4 value.
**if** $t_0 = 0$ **then**
    **for** $k$-th quantized weight matrix $\mathbf{W}_k$ **in** $\theta$ **do**
        Compute $Q(\mathbf{W}_k)$ via FP4 quant-dequant
        Compute sensitivity score: $s_k = |\mathbf{W}_k - Q(\mathbf{W}_k)|/\text{binwidth}(Q(\mathbf{W}_k))$
        Select $\text{idx}_k$ as the indices of the top-$p_{\text{sample}}\%$ elements in $s_k$
        Initialize $\widetilde{\mathbf{D}}_{M,k} \leftarrow \mathbf{0}, \widetilde{\mathbf{D}}_{Q,k} \leftarrow \mathbf{0}$
        Record snapshots: $\widetilde{\mathbf{W}'}_{M,k} \leftarrow \mathbf{W}_k[\text{idx}_k], \widetilde{\mathbf{W}'}_{Q,k} \leftarrow Q(\mathbf{W}_k)[\text{idx}_k]$
    **end for**
**else**
    **for** $k$-th quantized weight matrix $\mathbf{W}_k$ **in** $\theta$ **do**
        Compute $Q(\mathbf{W}_k)$ via FP4 quant-dequant
        Extract tracked subset: $\widetilde{\mathbf{W}}_{M,k} \leftarrow \mathbf{W}_k[\text{idx}_k], \widetilde{\mathbf{W}}_{Q,k} \leftarrow Q(\mathbf{W}_k)[\text{idx}_k]$
        Update distances:
$$\widetilde{\mathbf{D}}_{M,k} \leftarrow \widetilde{\mathbf{D}}_{M,k} + |\widetilde{\mathbf{W}}_{M,k} - \widetilde{\mathbf{W}'}_{M,k}|$$
$$\widetilde{\mathbf{D}}_{Q,k} \leftarrow \widetilde{\mathbf{D}}_{Q,k} + |\widetilde{\mathbf{W}}_{Q,k} - \widetilde{\mathbf{W}'}_{Q,k}|$$
        Update snapshots: $\widetilde{\mathbf{W}'}_{M,k} \leftarrow \widetilde{\mathbf{W}}_{M,k}, \widetilde{\mathbf{W}'}_{Q,k} \leftarrow \widetilde{\mathbf{W}}_{Q,k}$
    **end for**
**end if**
**Return:** $\text{idx}, \widetilde{\mathbf{D}}_M, \widetilde{\mathbf{D}}_Q$

---

---

**Algorithm 4** Trainer with Periodic Oscillation Suppression through Resetting (**OsciReset**)

---

**Input:** $\theta$: initialized model parameters; $\mathcal{D}$: batched data;

      $T_{\max}$: total number of training steps; $T_{\text{start}}$: suppression start step;

      $T_{\text{period}}$: suppression period length; $T_{\text{accu}}$: steps for oscillation detection;

      $\tau_{\text{osci}}$: oscillation risk threshold;

      $p_{\text{sample}}$ (optional): sampling ratio for memory-efficient statistics.

**Output:** Updated parameters $\theta_T$

**for** $t \leftarrow 1$ **to** $T_{\max}$ **do**

    Compute batch loss: $\mathcal{L}(\theta, \mathcal{D}_i)$

    Compute gradients: $\nabla_\theta \mathcal{L}$

    Update parameters: $\theta \leftarrow \text{OptimizerStep}(\theta, \nabla_\theta)$

    **if** $t \geq T_{\text{start}}$ **then**

      **if** $t \bmod T_{\text{period}} \leq T_{\text{accu}}$ **then**

        $\text{dist}_M, \text{dist}_Q \leftarrow \text{UpdateOscillationStats}(\theta, t \bmod T_{\text{period}})$

        **or** $\text{UpdateOscillationStats\_MemoryEfficient}(\theta, t \bmod T_{\text{period}}, p_{\text{sample}})$

      **end if**

      **if** $t \bmod T_{\text{period}} = T_{\text{accu}} + 1$ **then**

        $\text{OscillationSuppress}(\theta, \text{dist}_M, \text{dist}_Q, \tau_{\text{osci}})$  // OscillationSuppress is implemented in Alg. 1

      **end if**

    **end if**

**end for**

**Return:** $\theta$.

---

### C.2. Overhead & Memory Efficiency Analysis

**Overhead of Resetting Weights**   We only do Alg. 1 every $T_{\text{period}} = 200$ steps, which adds only $\sim 0.4$s per 200 steps of 150M-OLMo2 training (on $8\times$ RTX5090), less than $0.06\%$ of the total training time.

**Memory Efficiency**   In our 150M model, OsciReset introduces only around $3\%$ memory overhead when tracking all parameters using Alg. 2. For larger-scale training, we further provide a memory-efficient variant Alg. 3 that tracks only the top-5% weights closest to quantization thresholds (adding $\sim 0.6$ Bytes per parameter). The key intuition is that weights far from quantization decision boundaries rarely switch quantization bins, and therefore rarely oscillate.

As shown in Tab. 7, restricting tracking to near-threshold parameters preserves the statistical behavior of oscillation while substantially reducing memory cost. Empirically, this approximation has negligible impact on model quality. In distributed training, these tracking states are naturally sharded across GPUs, further reducing per-device memory overhead without additional communication cost.

*Table 7.* Memory usage and validation perplexity (PPL) of different implementations.

| | Track Ratio | Mem (Bytes/param) | 150M-Val. PPL |
|---|---|---|---|
| w/o OsciReset | - | - | 35.54 |
| Naive | 100% | $\sim 6.0$ | 34.95 |
| Memory-efficient | 5% | $\sim 0.6$ | 35.03 |

### C.3. Discussion on Oscillation-Quantization Error Relationship

We further examine the relationship between the harmful oscillators (denoted as set $A$) and weights with high quantization errors (i.e., those far from the quantization-bin centers, denoted as set $B$) in the final stage of training OLMo-2-150M, with the goal of understanding their overlap and distinguishing their roles in optimization.

**(1) Oscillators largely exhibit high quantization error** ($A \subset B$)**.** We analyzed the $< 5\%$ identified harmful oscillators and found that $\sim 70\%$ of them are located far from the quantization-bin centers (distance $> 0.8 \times$ bin radius). This observation validates the rationale behind OsciReset: by resetting the master weight to perfectly align with the quantized weight, we

simultaneously (1) eliminate their high quantization errors, (2) instantly mitigate the oscillation risk, and (3) provide them with a new, effective optimization trajectory.

This statistical insight also explains why our memory-efficient implementation Alg. 3 works well, because oscillating weights are mostly far from the bin-center. Moreover, since OsciReset is applied periodically (e.g., every ~200 steps), even if a few oscillating weights are missed in the current step, they will be captured and suppressed in subsequent cycles. This ensures the method maintains full efficacy with minimal overhead.

**(2) High quantization error does not imply oscillation ($B \not\subset A$).** Among all weights with high quantization errors (distance $> 0.8\times$ radius), only $\sim 15\%$ exhibit dynamic oscillation. The remaining weights are either relatively stable (e.g., their quantized values are already determined in later training stages) or possess determined gradients consistently updating in one direction, rather than oscillating back and forth. Therefore, simply resetting based on quantization error would disrupt normal optimization. Explicitly identifying the oscillating subset is necessary to target only the truly "harmful" weights.

## D. Detailed Results for Ablation Studies

### D.1. NVFP4 Linear Layer Design

**The Scaling Style of NVFP4 Quantization**   As described in Sec. 2, we set an outer-block scaling outside `NVFP4`'s micro-block to satisfy the numerical constraints, and our finer outer-block is more accurate; we adopt $1 \times 16$ group shape for $\mathbf{W}$ rather than $16 \times 16$ in (NVIDIA et al., 2025). In Tab. 8a, we reveal that both of our settings are better. We find that the per-tensor outer scaling would be detrimental, while the $1 \times 128$ outer scaling or per-row scaling would be similarly more accurate. Furthermore, $1 \times 16$ group shape outperforms $16 \times 16$ for weight quantization.

**The Unbiasedness of Gradient Estimation**   In Tab. 8b, we demonstrate the effectiveness of our more refined designs in unbiased backpropagation of `NVFP4` linear layer compared to NVIDIA et al. (2025). We surpass NVIDIA et al. (2025) through the alignment of $\widehat{\mathbf{X}}$ in forward/backward and the stochastic rounding, as described in Sec. 2.

**The Effect of Hadamard Transform**   We find that applying Hadamard Transformation (HT) for forward is harmful, while for the backward is beneficial. Moreover, while NVIDIA et al. (2025) suggests RHT brings benefits only for $d\mathbf{W}$, we see in Tab. 8c that under our unbiased linear framework, the Random Hadamard Transformation (RHT) would improve both computations of $d\mathbf{X}$ and $d\mathbf{W}$.

In conclusion, our `NVFP4` linear surpasses NVIDIA's `NVFP4` design (NVIDIA et al., 2025) with a more refined scaling method and unbiased gradient estimation as suggested in Sec. 2.

*Table 8.* Ablation study on `OLMo2-150M` model for `NVFP4` linear layer design. We reported training PPL averaged on the last 50 steps. (a) Ablation on block-size settings trained for `52B` tokens based on  TetraJet-v2-base. (b) Ablation on the unbiasedness of gradients trained for `52B` tokens. (StoQ.: stochastic quantization in $d\mathbf{W}$ compute) (c) Ablation on Hadamard Transform trained for `107B` tokens. (Fwd.: adopt HT in forward; dX/dW: adopt RHT in dX/dW computation of backpropagation)

<table>
<tr><td align="center" colspan="2">(a)</td><td align="center" colspan="2">(b)</td><td align="center" colspan="4">(c)</td></tr>
</table>

| Outer-Block Size | PPL |
|---|---|
| Per-Tensor | 31.75 |
| Per-Row | 31.58 |
| $1 \times 128$ | **31.49** |

| Weight-Inner-Block Size | PPL |
|---|---|
| $16 \times 16$ | 31.74 |
| $1 \times 16$ | **31.49** |

| Method | PPL |
|---|---|
| NVIDIA | 32.42 |
| w/o $\widehat{\mathbf{X}}$ align and StoQ. | 31.80 |
| w/o $\widehat{\mathbf{X}}$ align | 31.78 |
| TetraJet-v2-base | **31.49** |

| Fwd. | dX | dW | PPL |
|---|---|---|---|
| ✓ | ✗ | ✗ | 28.89 |
| ✗ | ✗ | ✗ | 28.67 |
| ✗ | ✗ | ✓ | 28.47 |
| ✗ | ✓ | ✓ | **28.42** |

### D.2. Oscillation Suppression and Outlier Selection

**The Effectiveness of Oscillation Suppression**   To prove our method OsciReset truly reduced the oscillation, we compute the metric $\mathrm{OsciRisk}(w)$ defined in Sec. 3.2 to identify the proportion of oscillation weights throughout the training process. In Fig. 7, there would be a drastic increase of oscillation weights if we do not suppress them, and we can observe a substantial decay to the oscillation when applying OsciReset, which confirms the effect of our method.

**Insensitivity to OsciReset Hyperparameters**  We further examine the sensitivity of OsciReset to two practical hyperparameters on OLMo-2-150M trained with 107B tokens: the oscillation threshold $\tau_{\mathrm{osci}}$ and the suppression starting point $T_{\mathrm{start}}$. As shown in Tab. 9 and 10, OsciReset is generally insensitive to these choices. Specifically, we uniformly set $\tau_{\mathrm{osci}} = 8$ and $T_{\mathrm{start}} \approx 60\% \cdot T_{\mathrm{max}}$ across all model sizes (70M to 370M) and data scales in our experiments. This consistency demonstrates the strong scalability and robustness of this empirical choice.

*Table 9.* Insensitivity to oscillation identifying threshold $\tau_{\mathrm{osci}}$ on OLMo-2-150M with 107B tokens of training.

| $\tau_{\mathrm{osci}}$ | Baseline (w/o OsciReset) | 4 | 8 | 12 | 16 |
|---|---|---|---|---|---|
| Val. PPL | 35.54 | 34.99 | **34.95** | 35.00 | 35.01 |

*Table 10.* Insensitivity to suppression starting point $T_{\mathrm{start}}$ on OLMo-2-150M with 107B tokens of training.

| $T_{\mathrm{start}}$ (Tokens) | No Suppression | ~40B | ~60B | ~80B |
|---|---|---|---|---|
| Val. PPL | 35.54 | 34.99 | **34.95** | 35.08 |

**Ablations on Outlier Selection**  In Tab. 11, we show that our static choice of the channels is effective by comparing it with randomly selecting channels to retain precision. Besides, our outlier controlling method improves both the forward and backward pass, which ensures the improvement in fully `NVFP4` training with forward and backward both quantized.

*Figure 7.* Change of oscillation proportion with/without OsciReset on `OLMo2-150M`. A weight $w$ is counted as oscillating if $\mathrm{OsciRisk}(w) > 16$.

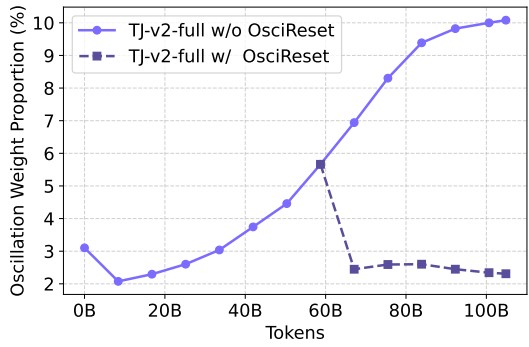

*Table 11.* Ablation on `150M` model for OutControl (Fwd./Bwd.: applied in forward/backward).

| Fwd. | Bwd. | PPL |
|---|---|---|
| ✗ | ✗ | 31.49 |
| ✓ | ✗ | 31.41 |
| ✓ | ✓ | **31.28** |

| Selection Style | PPL |
|---|---|
| None | 31.49 |
| Random | 31.47 |
| Largest norms | **31.28** |

# E. More FP4 Baselines Results

To better contextualize our method relative to recent low-bit training methods, we expand the evaluation to include three additional recent SOTA baselines (Wang et al., 2025; Tseng et al., 2025; Cook et al., 2026). For fairness, we adopt fully-FP4 settings whenever applicable. Tab. 12 shows that TetraJet-v2 consistently outperforms all compared methods, establishing a stronger empirical distinction from existing approaches.

*Table 12.* Expanded comparison on validation perplexity. We adopt fully-FP4 settings for fairness whenever applicable. Lower is better.

| Methods | 70M (52B tokens) | 150M (107B tokens) |
|---|---|---|
| BF16 | 45.27 | 33.49 |
| Wang et al. (2025) (FP4) | 97.22 | 82.96 |
| Tseng et al. (2025) (MXFP4) | 51.17 | 37.18 |
| FourOverSix (Cook et al., 2026) (NVFP4) | 50.99 | 36.65 |
| Quartet (MXFP4) | 51.23 | 36.89 |
| NVIDIA (NVFP4) | 50.94 | 36.73 |
| TetraJet-v2-base (Ours, NVFP4) | 49.33 | 35.88 |
| TetraJet-v2-full (Ours, NVFP4) | **47.75** | **34.95** |

## F. Compatibility with Muon

We additionally evaluate TetraJet-v2 with Muon to verify that our method is not tied to a particular optimizer. This is important because TetraJet-v2 operates at the quantization level, and its effect should therefore be largely orthogonal to the optimizer choice. Tab. 13 confirms this expectation. TetraJet-v2 consistently improves over the NVIDIA baseline under Muon, with TetraJet-v2-full achieving the best validation perplexity among all 4-bit methods.

*Table 13.* Performance with Muon on OLMo-2-70M and OLMo-2-150M trained with 52B tokens. Lower is better.

| Method | 70M-Val. PPL | 150M-Val. PPL |
|---|---|---|
| NVIDIA | 50.77 | 40.67 |
| TetraJet-v2-base (Ours) | 49.81 | 39.63 |
| TetraJet-v2-full (Ours) | **47.57** | **38.73** |
| BF16 | 45.11 | 37.56 |

## G. Detailed Efficiency Results for TetraJet-v2

### G.1. Latency Overhead of Finer-Grained Weight Scaling Scheme (1×16 vs 16×16)

We report the end-to-end overhead of our $1\times16$ weight scaling scheme on a single-layer Transformer block based on TetraJet-v2-base on RTX 5090. Compared with the $16\times16$ baseline, the average overhead is only 2.49% across all evaluated hidden sizes and sequence lengths (Tab. 14). This is a controlled comparison that changes only the weight quantization recipe; moreover, the $16\times16$ baseline is particularly efficient since its backward pass does not require weight re-quantization. Nevertheless, our finer-grained scheme adds only minimal overhead.

*Table 14.* Transformer block End-to-end overhead ratio (%) against the $16\times16$ recipe on RTX 5090 (`MicroBatchSize=8`).

| Hidden Size \ SeqLen | 1024 | 2048 | 4096 |
|---|---|---|---|
| 2048 | 0.98% | 1.11% | 1.01% |
| 4096 | 3.10% | 2.91% | 2.16% |
| 8192 | 4.48% | 3.87% | 2.83% |

### G.2. Speedup of a Single Linear Layer

To validate the performance of our linear layer and overhead, we test the throughput on a single linear layer. It is noteworthy that these performance gains are achieved even when accounting for the full overhead, including quantization/de-quantization for all methods and the Random Hadamard Transform specifically required by the TetraJet-v2 variants. As shown in Fig. 8, TetraJet-v2-full exhibits slightly lower throughput than the base version due to its mix-precision design ($p = 10\%$ activation channels are turned into `FP8` in both forward and backward), yet it still maintains a substantial lead over the FP8 baseline.

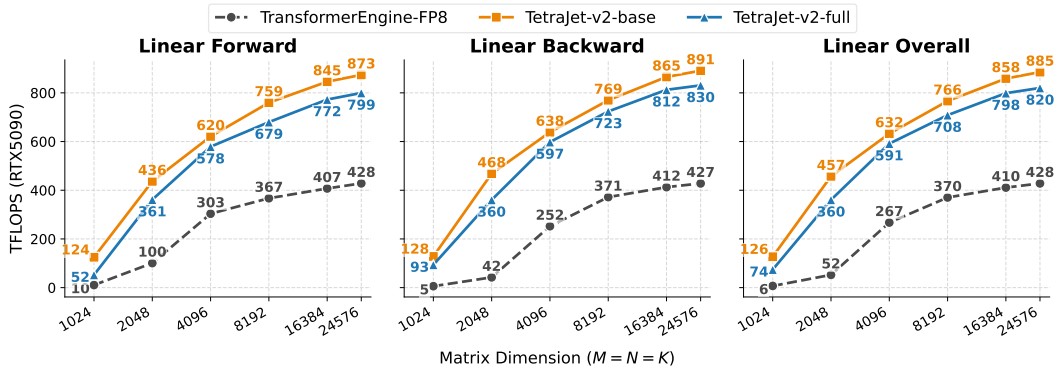

*Figure 8.* Computational throughput (TFLOPS) of a Linear layer across various matrix dimensions ($M = N = K$) on RTX 5090 (`CUDA Version=12.8`). All benchmarks account for the *full overhead*, like quantization, de-quantization, and Hadamard Transform.

### G.3. End-to-end Speedup on a Complete Transformer Layer

We implement a full Transformer block with FlashAttention-2 (Dao, 2024) and apply efficient RMSNorm (Zhang & Sennrich, 2019) implementation from TransformerEngine (TE) (NVIDIA, 2024). We fix `MicroBatchSize = 4` and `SeqLen = 1024`. In Fig. 9, as the `HiddenSize` scales from 4096 to 16384, the proportion of time consumed by linear layers increases significantly. Since our methods primarily accelerate the linear operations, the overall speedup becomes more pronounced in larger models. Specifically, at Hidden Size = 16384, TetraJet-v2-base (TJ-v2-base) achieved a $1.75\times$ speedup. Even with the accuracy retained by keeping $p = 10\%$ activation channels in `FP8`, TetraJet-v2-full (TJ-v2-full) still provides a $1.67\times$ speedup over the baseline, demonstrating the robustness of our approach for large-scale Transformer architectures.

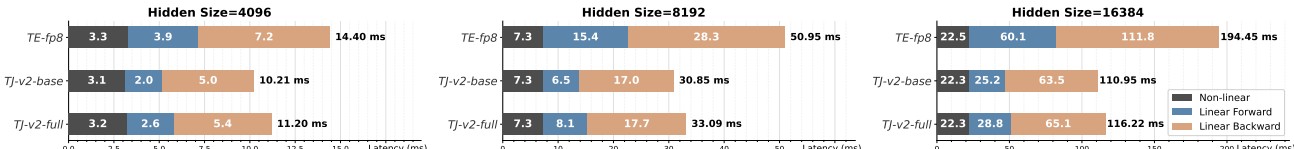

*Figure 9.* Latency breakdown of a complete Transformer block on RTX 5090 (`CUDA Version=12.8`) across various Hidden Sizes (`MicroBatchSize = 4` and `SeqLen = 1024`). The total latency is decomposed into Non-linear (including FlashAttention-2, RMSNorm, and activations), Linear Forward, and Linear Backward components.

We provide more detailed results on more `MicroBatchSize` and `SeqLen` in Tab. 15.

*Table 15.* End-to-end speedup ratio of a complete Transformer block against TE-FP8 under different `MicroBatchSize`(BS) and `SeqLen`.

*(a)* Hidden Size= 8192

| SeqLen | BS=2 | | BS=4 | | BS=8 | |
|---|---|---|---|---|---|---|
| | 2048 | 4096 | 2048 | 4096 | 2048 | 4096 |
| TJ-v2-base | 1.58× | 1.49× | 1.53× | 1.49× | 1.53× | 1.49× |
| TJ-v2-full | 1.48× | 1.40× | 1.44× | 1.41× | 1.43× | 1.39× |

*(b)* Hidden Size= 16384

| SeqLen | BS=1 | | BS=2 | | BS=3 | |
|---|---|---|---|---|---|---|
| | 1024 | 2048 | 1024 | 2048 | 1024 | 2048 |
| TJ-v2-base | 1.87× | 1.74× | 1.83× | 1.67× | 1.75× | 1.66× |
| TJ-v2-full | 1.79× | 1.66× | 1.73× | 1.59× | 1.66× | 1.57× |

### G.4. Full-Model End-to-End Speedup

To better assess the practical system-level benefit, we further report the end-to-end speedup on *full Llama2-structured models* against TransformerEngine-FP8 (TE-FP8). These results in Tab. 16 show that our method preserves clear end-to-end speedup beyond single-layer microbenchmarks, at the scale of full models.

We also note that truly comprehensive system-level acceleration depends on many other framework components, such as communication, data loading, runtime scheduling, optimizer states, and other overheads. Our current work focuses on accelerating the model's **forward** and **backward** computation, so we report the corresponding forward/backward end-to-end speedup here. We hope future high-performance systems research can further optimize the remaining framework components and address the full-stack acceleration problem more comprehensively.

*Table 16.* End-to-end speedup against TransformerEngine-FP8 (TE-FP8) on full Llama2-structured models.

| Model | Device | Ours-base Fwd | Ours-base Bwd | Ours-full Fwd | Ours-full Bwd |
|---|---|---|---|---|---|
| 2B | RTX 5090 | 1.58× | 1.65× | 1.39× | 1.61× |
| 7B | RTX 6000 Pro | 1.82× | 1.67× | 1.53× | 1.60× |

