# OpenReview forum: "TetraJet-v2: Accurate NVFP4 Training for Large Language Models with Oscillation Suppression and Outlier Control"
_ICML.cc/2026/Conference — ICML 2026 spotlight_

### Official Review · Reviewer_kwnm · 2026-02-19

**Soundness:** 3
**Presentation:** 3
**Significance:** 3
**Originality:** 2
**Overall Recommendation:** 4
**Confidence:** 4

**Summary:**

TetraJet-v2 proposes a fully 4-bit (NVFP4) training framework for large language models, explicitly targeting optimization instabilities that arise in ultra-low-precision training. An important topic analyzed by the manuscript is weight oscillation and activation outliers, which the authors identify as primary contributors to the performance gap between FP4 and BF16 training. The method introduces three key components: an unbiased double-block quantization scheme for NVFP4 linear layers, OsciReset to suppress oscillating weights, and OutControl to manage structured activation outliers.

Overall, the authors outline a key concept that optimization-aware stabilization techniques—beyond reducing quantization error alone—are crucial for enabling accurate and efficient end-to-end FP4 LLM pretraining. Experiments across models with up to 370M parameters trained on up to 212B tokens demonstrate improved perplexity compared to prior FP4 baselines.

**Compliance With Llm Reviewing Policy:**

Affirmed.

**Final Justification:**

The authors have satisfactorily addressed my main concerns in the rebuttal, I have updated my recommendation from weak reject to weak accept.

**Key Questions For Authors:**

1.	The authors adopt a more fine-grained weight block shape (1 × 16) instead of the 16 × 16 grouping suggested by NVIDIA. While this design likely improves quantization accuracy, it would be helpful to further discuss the potential computational or implementation overhead introduced by this choice.

2.	To address the weight oscillation issue, the authors propose resetting the master weights to the centers of their corresponding quantization bins (OsciReset). While this strategy effectively reduces oscillation and mitigates quantization error, it may also introduce additional perturbations due to the reset operation itself. It would be valuable for the authors to further clarify how the trade-off between reset-induced error and quantization error is balanced in practice, and whether there are theoretical or empirical guarantees supporting this balance.

3.	In Section 3.2, the authors propose tracking the optimization distance between master weights and their quantized counterparts to identify oscillating weights. This oscillation risk metric is simple and intuitively motivated, and appears effective in practice. However, maintaining per-weight statistics (e.g., distM and distQ) may introduce additional memory overhead. It would be helpful if the authors could clarify the associated memory cost and discuss whether this overhead is negligible compared to the overall model size, especially for larger-scale LLMs.

4.	The novelty and contribution of the proposed OutControl method appear somewhat limited. The approach largely resembles a straightforward application and extension of existing techniques, including Hadamard-based rotation methods (e.g., Xi et al., 2023; Tseng et al., 2025; Castro et al., 2025), high-precision retention for outlier channels as in LLM.int8() (Dettmers et al., 2022), and NVIDIA’s NVFP4 training recipe (NVIDIA et al., 2025). While the empirical results are promising, the paper would benefit from a clearer articulation of the conceptual distinctions and technical innovations that differentiate OutControl from these prior approaches.

5.	In Table 2, the authors compare their method only with Quartet and NVIDIA’s FP4 training recipe. The empirical evaluation would be strengthened by including comparisons with other closely related FP4 training approaches, such as Tseng et al. (2025) and Wang et al. (2025). Incorporating these baselines would provide a more comprehensive and balanced assessment of the relative advantages of the proposed method.

6.	The experimental evaluation primarily reports results on classification-style benchmarks (as shown in Table 2). It would strengthen the paper to additionally evaluate performance on generation-oriented tasks, such as GSM8K, which better reflect reasoning capabilities. Including such benchmarks would provide a more comprehensive assessment of the proposed method’s effectiveness in practical LLM deployment scenarios.

7.	In Section 5.3, the authors report the end-to-end speedup at the level of a single Transformer layer. It would be helpful to also provide the overall training or inference speedup for the full model, in order to better assess the practical system-level benefits of the proposed method.

**Limitations:**

The limitations and societal impact discussion is currently very brief and lacks concrete analysis. It would be helpful for the authors to more explicitly discuss potential broader impacts of enabling more efficient large-scale LLM training, including both positive and negative implications.

**Strengths And Weaknesses:**

Strengths:

The paper is well written and easy to follow. The authors clearly identify a critical challenge in low-precision training—namely, weight oscillation—and articulate its impact on optimization stability. The proposed OsciReset method is simple yet effective, and is well supported by both intuitive explanation and empirical evidence.

Weaknesses:

While the paper proposes targeted techniques such as unbiased double-block quantization, OsciReset, and OutControl, some components—especially OutControl—appear incremental relative to existing rotation-based and outlier-retention methods. Overall, the authors outline a central concept that optimization-aware stabilization is essential for accurate ultra-low precision training; however, the empirical comparison is somewhat limited, as only a small number of FP4 baselines are included, making it harder to fully assess the relative novelty and advantage of the proposed approach .

---

> ### Author Rebuttal · Authors · 2026-03-30
>
> We sincerely thank the reviewer for the professional feedback. We have provided **three** additional recent FP4 training baselines, along with comprehensive analysis and clarifications to address concerns.
>
> > Question 1 (Q1):
> > Latency overhead of weight-quant (1×16 vs. 16×16)
>
> Our finer-grained 1×16 scaling introduces **negligible 2.49% latency overhead** on average compared to the 16×16 baseline (across hidden in {2048,4096,8192}, and SeqLen in {1024,2048,4096}), making accuracy improvement highly worthwhile.
>
> > Q2:
> > Balance of Reset-induced vs. Quantization Error
>
> Decoupling these two errors analytically is challenging, but we clarify why OsciReset maintains optimization integrity through two key aspects:
> - We only target the **<5%** of weights identified as "harmful" oscillators; by OsciReset, we allow them to find new optimization **without altering the current quantized weight**.
> - The vast majority of weights (**>95%**) remain entirely untouched, ensuring no additional perturbations.
>
> > Q3:
> > Memory overhead of identifying oscillating weights
>
> In our 150M baseline, OsciReset adds only ~3% memory overhead.
>
> For **larger-scale**, we provide **memory-efficient implementation** by tracking only the **top-5%** weights closest to thresholds (adds **~0.6 Bytes/param**). This maintains efficacy because weights far from thresholds rarely oscillate. In distributed training, the states are naturally **sharded** across GPUs, further reducing the per-device memory footprint.
>
> *Table 1: Memory usage and PPL of implementations.*
> ||Track Ratio| Mem (Bytes/Param) | 150M-Val.PPL |
> |-|-|-|-|
> | w/o OsciReset|-|-|35.54|
> | Naive| 100%| ~6.0| 34.95|
> | Mem. Efficient | **5%**| ~**0.6**| 35.03|
>
> > Q4 & Weaknesses 1 (W1):
> > OutControl appears incremental to existing techniques. What are its distinctions?
>
> We clarify our novelty and contributions beyond incremental integration as follows:
>
> 1. **Non-trivial Optimal Recipe for LLMs:**  While Outlier-Control integrates existing techniques, our contribution here is **establishing the optimal practical recipe** for fully-FP4 training. Our rigorous evaluation on a massive data scale (**500~600x** token-to-parameter ratio) makes optimization significantly harder and provides a reliable benchmark. In this challenging setting, **correctly** applying these insights, such as fixing activation misalignment for strictly unbiased backward, is critical for long-term convergence. Meanwhile, we apply outlier retention to **both forward and backward**, while previous works focus only on the forward.
> 2. **Massive Empirical Gains:** Simply by implementing our optimal unbiased-backward and RHT settings (*even before adding OsciReset and mix-precision*), `TetraJet-v2-base` reduces Val. PPL from 26.2 (NVIDIA’s recipe) to 25.5, **closing PPL-gap-to-BF16 by 28.0%**. With all components integrated, we reduce the gap by an **average of 51.3%** over current SOTA methods.
>
> We also emphasize that our **primary algorithmic innovation** is **OsciReset**, the **first** method addressing LLM weight oscillation, a challenge orthogonal to outliers and overlooked in prior work. Unlike ViT-based techniques that fail to generalize to LLMs, OsciReset is simple yet effective and ensures stability with little overhead.
>
> > Q5 & W2:
> > More FP4 Baselines
>
> We expand evaluation by including **three** more SOTA baselines [1,2,3]. Results demonstrate that TetraJet-v2(Ours) outperforms all existing 4-bit methods.
>
> *Table 2: Expanded comparison. We adopt **fully**-FP4 settings for fairness. [1] targets forward-only quantized training and does not generalize to fully-FP4 setting.*
>
> |Methods|70M-PPL|150M-PPL|
> |-|-|-|
> |BF16|45.27|33.49|
> |**[1]**(FP4)|97.22|82.96|
> |**[2]**(MXFP4)|51.17|37.18|
> |**[3]**(NVFP4)|50.99|36.65|
> |Quartet|51.23|36.89|
> |NVIDIA|50.94|36.73|
> |Ours-base|49.33|35.88|
> |Ours-full|**47.75**|**34.95**|
>
> > Q6:
> > Performance on Generation Tasks (GSM8K)
>
> Since our current pre-trained model/data-size are relatively small, we fine-tuned Llama-3.2-1B for 700 steps.
>
> *Table 3: GSM8k fine-tuning on Llama-3.2-1B.*
> |Method|Acc%(Mean±Std)|
> |-|-|
> |NVIDIA|23.39±0.72|
> |Ours-base|23.73±0.80|
> |Ours-full|**24.37**±1.14|
> |BF16|25.83±0.85|
>
> > Q7:
> > Full Model Speedup
>
> We report the overall end-to-end speedup against TransformerEngine-FP8 (TE-FP8) on **full** Llama2-structured models:
> - **2B (RTX5090):** Ours-base (1.58x Fwd, 1.65x Bwd) | Ours-full (1.39x Fwd, 1.61x Bwd)
> - **7B (RTX6000 Pro):** Ours-base (1.82x Fwd, 1.67x Bwd) | Ours-full (1.53x Fwd, 1.60x Bwd)
>
> > Limitations: more discussion on broader impacts
>
> Thanks for the valuable suggestion! We will expand in the revision. Positive impacts include democratizing LLM pre-training and reducing carbon footprint through 4-bit efficiency. Negative implications involve lowered barriers for automated misinformation.
>
> *[1] Wang et al., "Optim... FP4 Quantization," ICML 2025*
> *[2] Tseng et al., "Training LLMs with MXFP4," AISTATS 2025*
> *[3] Cook et al., "Four Over Six ...," 2025*

---

> > ### Author Rebuttal · Reviewer_kwnm · 2026-04-03
> >
> > The authors have addressed most of my concerns, and I will raise my score accordingly. I have one remaining question regarding the balance between reset-induced effects and quantization error. In particular, could it be that the 5% of weights identified as harmful oscillators largely overlap with those that incur high quantization errors?

---

> > > ### Author Response · Authors · 2026-04-03
> > >
> > > Dear Reviewer kwnm,
> > >
> > > We sincerely thank you for taking the time to review our rebuttal and for your positive and constructive feedback. We deeply appreciate your insightful follow-up question.
> > >
> > > We would like to clarify the relationship between the harmful oscillators (Set $A$) and weights with high quantization errors (Set $B$) through two key observations:
> > >
> > > - **1. Oscillators largely have high quantization errors** ($A\subset B$):
> > >   We analyzed the <5% identified harmful oscillators and found that ~70% of them are located far from the quantization-bin centers (distance > 0.8 $\times$ bin radius). This observation validates the rationale behind **OsciReset**: by resetting the master weight to perfectly align with the quantized weight, we simultaneously (1) eliminate their high quantization errors, (2) instantly mitigate the oscillation risk, and (3) provide them with a new, effective optimization trajectory.
> > >
> > >   *Additional note on the memory-efficient implementation:*
> > >   This statistical insight also explains why our memory-efficient implementation (tracking only the top-5% weights closest to thresholds) works well, because oscillating weights are mostly far from the bin-center. Moreover, since OsciReset is applied periodically (e.g., every ~200 steps), even if a few oscillating weights are missed in the current step, they will be captured and suppressed in subsequent cycles. This ensures the method maintains full efficacy (as supported by Rebuttal Table 1) with minimal overhead.
> > >
> > > - **2. However, not all high-error weights are oscillating ( $B\not\subset A$ ):**
> > >   Among all weights with high quantization errors (distance > 0.8 $\times$ radius), only **~15%** exhibit dynamic oscillation. The remaining weights are either relatively stable (e.g., their quantized values are already determined in later training stages) or possess determined gradients consistently updating in one direction, rather than oscillating back and forth. Therefore, simply resetting based on quantization error would disrupt normal optimization. Explicitly *identifying* the oscillating subset is strictly necessary to target only the truly "harmful" weights.
> > >
> > > We will include these statistics and discussions on the reset-error balance in our revision. Once again, thank you for your dedicated time and valuable feedback that helped improve our paper.
> > >
> > > Best regards,
> > > The Authors

---

### Official Review · Reviewer_gk34 · 2026-03-10

**Soundness:** 3
**Presentation:** 3
**Significance:** 2
**Originality:** 2
**Overall Recommendation:** 4
**Confidence:** 5

**Summary:**

This paper presents TetraJet-v2, an end-to-end 4-bit fully-quantized training (FQT) method that aims to reduce the accuracy gap between 4-bit training and BF16 training for LLMs. The authors attribute the main optimization difficulties in 4-bit training to weight oscillation and activation outliers. To address these issues, they propose: (1) an unbiased double-block quantization scheme for NVFP4 linear layers, (2) OsciReset, which suppresses oscillating weights by resetting them to quantization-bin centers, and (3) OutControl, which keeps persistent outlier channels in higher precision to protect accuracy.

**Compliance With Llm Reviewing Policy:**

Affirmed.

**Final Justification:**

My concerns have been addressed

**Key Questions For Authors:**

1. Per-block (128) global scale requires storing more scales than per-tensor scaling. Why is per-block (128) scaling considered more hardware-friendly than per-tensor scaling in your implementation/runtime?
2. For “Identifying Oscillating Weights”, what is the additional memory cost (e.g., bytes/parameter or % of total training memory)? Please include whether extra tensors are stored per weight, per block, or per layer.
3. What is the execution frequency of Algorithm 1 (every step, every N steps, per epoch, or only at specific phases)? What is its measured runtime overhead?

**Limitations:**

Yes

**Strengths And Weaknesses:**

## Strength
1.  The paper provides clear and thorough ablation studies on quantized linear-layer settings, which are easy to follow.
2. The method achieves meaningful end-to-end speedups.
## Weakness
1.  Prior work (e.g., [1,2]) offers analysis on why random Hadamard rotation can sometimes hurt performance. Including and discussing these results could strengthen the experimental section and interpretation.
2. The “Identifying Oscillating Weights” step may increase memory usage due to extra optimizer-state or tracking information, which is a major concern for LLM-scale training.
3. The method appears largely as a careful ablation and combination of existing techniques. For example, TetraJet-v2-base seems to combine known components such as finer-grained scaling, double quantization, stochastic rounding, and Hadamard rotation. It would help to clarify what is new beyond engineering integration and tuning.
4. The paper mentions that oscillation-mitigation methods from ViT quantized training do not transfer well to LLMs. The reason is unclear: is it mainly due to architecture differences, data/optimization behavior, or 4-bit precision constraints?
[1] Bridging the Gap Between Promise and Performance for Microscaling FP4 Quantization
[2]  INT v.s. FP: A Comprehensive Study of Fine-Grained Low-bit Quantization Formats

---

> ### Author Rebuttal · Authors · 2026-03-30
>
> We sincerely thank the reviewer and provide following analysis and clarification to address the reviewer's professional concerns.
>
> > Weakness 1 (W1):
> > Not include prior work (e.g.,[1,2]) on Hadamard rotation
>
> [1,2] find rotation may degrade NVFP4 quality (e.g., lower QSNR), consistent with our result on forward HT. Our work further finds that RHT improves both dW and dX with unbiased gradients. We will add this discussion in the revision.
>
> > W2 & Question 2 (Q2):
> > For “Identifying Oscillating Weights”, what is the additional memory cost?
>
> Tracking states are stored **per weight element** within the quantized subset. In our 150M baseline, OsciReset adds only ~3% memory overhead when tracking the whole model.
>
> For **larger-scale**, we provide **memory-efficient implementation** by tracking only the **top-5%** weights closest to thresholds (adds **~0.6 Bytes/param**). We store an index along with four subset-tracking states (previous master/quantized weight and accumulated distance). This maintains efficacy because weights far from thresholds rarely oscillate. In distributed training, the states are naturally **sharded** across GPUs, further reducing the per-device memory footprint.
>
> *Table 1: Memory usage and PPL of implementations.*
> | |Track Ratio|Mem(Bytes/param)|150M-Val.PPL|
> |-|-|-|-|
> |w/o OsciReset|-|-|35.54|
> |Naive|100%|~6.0|34.95|
> |Mem. Efficient|**5%**|~**0.6**|35.03|
>
> > W3:
> > What is new beyond engineering integration and tuning?
>
> We clarify our novelty and contributions beyond engineering integration as follows:
>
> 1. **Core Algorithmic Innovation (OsciReset):** Our primary algorithmic innovation is **OsciReset**, the **first** method addressing LLM weight oscillation—a challenge orthogonal to outliers and overlooked in prior work. Unlike ViT-based techniques that fail to generalize to LLMs (see response to W4), OsciReset is a simple yet effective algorithm with negligible overhead (see response to Q2&Q3).
> 2. **Non-trivial Optimal Recipe for LLMs:** While some components are acknowledged, identifying the **practically optimal configuration** for LLMs is highly non-trivial. Our rigorous evaluation on a massive data scale (**500~600x** token-per-parameter ratio) makes optimization significantly hard and provides a reliable benchmark. In this challenging setting, **correctly** applying these insights, such as fixing activation misalignment to ensure unbiasedness, and extending outlier retention to **both forward and backward passes**—is critical for long-term convergence.
> 3. **Massive Empirical Gains:** Simply by implementing our optimal unbiased-backward and RHT settings (*even before adding OsciReset and mix-precision*), our `TetraJet-v2-base` reduces Val. PPL from 26.2 (NVIDIA’s recipe) to 25.5, **closing the PPL-gap-to-BF16 by 28.0%**. With all components integrated, we reduce the gap by an **average of 51.3%** over current SOTA methods.
>
> > W4:
> > Why oscillation-mitigation methods from ViT do not transfer well to LLMs?
>
> We clarify that this failure is primarily due to:
>
> 1. **Differences in data/optimization behavior**
>    Image samples only provide 1 class label, resulting in a sparse supervision density. A typical ViT-Base only reaches a Target-TPP (Tokens-Per-Parameter) of ~4.5 (ImageNet, 300 epochs). This under-trained setting allows models to fit even random labels [3].
>    LLMs (>500 TPP) are heavily trained by token-level targets. Therefore, oscillation-mitigation methods designed for image-classification ViTs may no longer be effective for maintaining LLM stability.
>
> 2. **Previous ViT methods have inherent flaws for long-term training**
>    Previous methods (e.g., Q-EMA, Freeze, Dampen) harm global optimization or blocks weights from being optimized further. As shown in paper Figure 4 (Page 7), their losses would rebound severely after only ~5B tokens.
>    Our OsciReset works by *explicitly* resetting specific oscillating weights' optimization trajectory **without hindering the normal loss decrease**.
>
> *Table 2: PPL of Oscillation-mitigations (OLMo-2-150M).*
>
> |Method|Val.PPL|
> |-|-|
> |No Suppression|35.54|
> |Q-EMA|36.40|
> |Freeze|36.00|
> |Dampen|36.33|
> |OsciReset(Ours)|**34.95**|
>
> *[3] Zhang et al., "Understanding deep learning requires rethinking generalization." ICLR 2017*
>
> > Q1:
> > Why is per-block (128) scale considered more hardware-friendly?
>
> We clarify that it is an implementation detail rather than a main contribution of our work.
>
> Per-tensor scaling needs a global reduction to get the full-tensor absmax, while outer-block (128) scaling can fuse local max into the quant kernel (avoiding a full extra scan). For larger-LLM future work, it is possible to explore larger granularity (e.g., per-row) to balance scale-overhead and runtime efficiency.
>
> > Q2: (We respond in W2)
>
> > Q3:
> > Execution frequency and runtime overhead of Algorithm 1?
>
> We only do Alg.1 every 200 steps, which adds only ~0.4s per 200 steps of 150M-OLMo2 training (on 8x RTX5090), less than 0.06% of the total training time.

---

> > ### Author Rebuttal · Reviewer_gk34 · 2026-04-03
> >
> > My concerns have been addressed, I increase my score from 3 to 4.

---

> > > ### Author Response · Authors · 2026-04-03
> > >
> > > Dear Reviewer gk34,
> > >
> > > Thank you for reviewing our rebuttal and for your positive feedback. We are glad that our responses have adequately addressed your concerns.
> > >
> > > We appreciate your constructive comments during the review process. We will incorporate the discussions and additional experiments provided in the rebuttal into the revised manuscript.
> > >
> > > Thank you again for your time and effort in reviewing our work.
> > >
> > > Best regards,
> > > The Authors

---

### Official Review · Reviewer_Lwpw · 2026-03-11

**Soundness:** 3
**Presentation:** 3
**Significance:** 3
**Originality:** 2
**Overall Recommendation:** 4
**Confidence:** 4

**Summary:**

This paper proposes TetraJet-v2, an end-to-end NVFP4 4-bit fully-quantized training method for LLMs that addresses the key bottlenecks of weight oscillation and activation/gradient outliers via an unbiased double-block quantization scheme, OsciReset and OutControl. Evaluated on OLMo2 models up to 370M parameters (212B tokens), TetraJet-v2 outperforms SOTA FP4 methods, reducing the performance gap to BF16 by 51.3% on average and delivering a 1.67× end-to-end speedup over FP8.

**Compliance With Llm Reviewing Policy:**

Affirmed.

**Final Justification:**

My concerns have been addressed

**Key Questions For Authors:**

1. Does the stochastic rounding in the backward pass cause training oscillations?
2. Have the authors compared with the Muon optimizer, which is widely adopted in LLM training? Can TetraJet-v2 still work well with the Muon optimizer?
3. Has the author performed an ablation study on starting the oscillation suppression at different training steps, and how much does this affect the final results?

**Limitations:**

yes

**Strengths And Weaknesses:**

## Strengths
1. The method designs LLM-tailored algorithms to address the core problems of weight oscillation and outliers with minimal computational overhead.
2. It has an optimized NVFP4 linear layer with unbiased gradient estimation, which offers a theoretical guarantee for SGD convergence.
3. Rigorous experiments and detailed analyses are carried out to verify the method’s efficacy and provide valuable insights for low-bit training research.
4. Hardware-friendly CUDA kernels are implemented, which well balance high model accuracy and remarkable training speedups.

## Weaknesses
1. Outlier Control is merely a combination of existing methods (RHT, mixed-precision), with limited novelty.
2. Weight quantization adopts a finer-grained scaling scheme (1×16 vs. 16×16). Only performance comparisons are provided, while the analysis of its impact on latency is missing. It is unclear whether such a trade-off is worthwhile.
3. Currently, all model sizes released by OLMo2 are above 1B parameters, but the authors only validate the method on a 370M model. The scale is too small, and it is uncertain whether comparable performance can be achieved on larger models.
4. The authors claim 2× lower memory usage than MXFP8, but no concrete experimental results are provided to support this statement.

---

> ### Author Rebuttal · Authors · 2026-03-30
>
> We greatly thank the reviewer for constructive suggestions. We respond to the questions below.
>
> > Weakness 1 (W1):
> > Outlier Control is merely a combination of existing methods (RHT, mixed-precision), with limited novelty.
>
> Thanks for the feedback. We **first clarify that our core algorithmic innovation is OsciReset**, the **first** method addressing weight-oscillation in low-precision LLMs—a crucial optimization challenge completely orthogonal to outlier control, which is largely overlooked in existing literature.
>
> While Outlier-Control integrates existing techniques, applying them *correctly* in LLM training is non-trivial. Our contribution here is **establishing the optimal practical recipe** for fully-FP4 training: (1) identify and fixing activation misalignment in NVIDIA's backward to ensure strictly unbiased gradient estimation; (2) extend outlier retention to the backward pass, improving gradient accuracy throughout training. Both target accurate gradient computation: the foundation of long-term convergence.
>
> The value of the recipe lies in its **massive empirical gains**: By simply implementing unbiased backward and optimal RHT (*before adding mix-precision or OsciReset*), our `TetraJet-v2-base` reduces the Val.PPL of 370M model from 26.2 (NVIDIA's) to 25.5, **closing the PPL-gap-to-BF16 by 28.0%**. Furthermore, integrating all components, we reduce the gap-to-BF16 by **an average of 51.3%** over current SOTA FP4 methods.
>
> > W2:
> > Latency overhead of a finer-grained weight scaling scheme (1×16 vs. 16×16).
>
> Thanks for the suggestion. We clarify that our finer-grained-weight 1×16 scaling scheme introduces negligible 2.49% latency overhead on average compared to the 16×16 baseline, making the accuracy improvement highly worthwhile.
>
> *Table 1: End-to-End Overhead Ratio against 16x16 recipe on RTX5090 (BS=8).*
>
> |Hidden Size \ SeqLen |1024|2048|4096|
> |-|-|-|-|
> |2048|0.98%|1.11%|1.01%|
> |4096|3.10%|2.91%|2.16%|
> |8192|4.48%|3.87%|2.83%|
>
> > W3:
> > the authors only validate on a small 370M model
>
> We acknowledge the computational constraints limiting our model size, but emphasize our evaluation on a massive data scale with a **token-to-parameter ratio of 500~600x** in the paper. Such data-size makes low-bit optimization harder, providing a highly reliable benchmark to evaluate training methods.
>
> To demonstrate scalability on model-size, we trained a MoE model **with total 2.2B parameters** with 100B tokens based on OLMo-2 structure. Due to time and resource limits, we only compared our TetraJet-v2-base against NVIDIA baseline.
>
> *Table 2: PPL on 2.2B MoE model (500M activated params).*
>
> | Method| Train PPL | Val. PPL  |
> |-|-|-|
> | NVIDIA |15.63|19.93|
> | TetraJet-v2-base (Ours) | **15.38** | **19.64** |
> | BF16 |14.87|19.00|
>
> >  W4:
> >  The authors claim 2× lower memory usage than MXFP8 ...
>
> We clarify that the mention of "2× lower memory usage" in our introduction refers to **the memory usage of NVFP4 data format itself** compared to MXFP8 [1], rather than an algorithmic claim of our paper. For concrete empirical validation, recent large-scale model deployments confirm that NVFP4 practically achieves ~1.5$\times$ to 1.8$\times$ smaller effective weight storage compared to FP8 [2].
>
> > Question 1 (Q1):
> > Does the stochastic rounding (SR) in the backward pass cause training oscillations?
>
> **No.** Theoretically, SR ensures unbiased gradient estimation and SGD convergence [3]. Its variance is strictly bounded and does not trigger systematic oscillations. Empirically, weight oscillation stems from latent weights hovering near quantization thresholds rather than the gradient estimation itself. This is confirmed in Figure 7 (Page 15), where oscillations are effectively suppressed by OsciReset even with SR active, and in Figure 3 (Page 7), which shows a stable loss decay curve.
>
> > Q2:
> > ... Can TetraJet-v2 still work well with the Muon optimizer?
>
> **Yes.** We confirm that TetraJet-v2 is compatible with Muon because our method operates at the quantization level, which is orthogonal to optimizing rules. Results are below.
>
> *Table 3: Performance with Muon on OLMo-2-70M and 150M with 52B tokens.*
>
> | Method | 70M-Val.PPL | 150M-Val.PPL |
> |-|-|-|
> | NVIDIA | 50.77 | 40.67 |
> | TetraJet-v2-base (Ours) | 49.81 | 39.63 |
> | TetraJet-v2-full (Ours) | **47.57** | **38.73** |
> | BF16 | 45.11 | 37.56 |
>
> > Q3:
> > Ablation study on starting the oscillation suppression at different training steps?
>
> **Yes.** We found that **the final results are insensitive** to the starting step.
>
> *Table 4: Impact of starting step for OsciReset on OLMo-2-150M with 107B tokens.*
>
> |$T_{start}$ (Tokens) | Val.PPL|
> |-|-|
> |No Suppression| 35.54 |
> |~40B|34.99|
> |~60B|**34.95**|
> |~80B|35.08|
>
> *[1] NVIDIA, "Pretraining ... with NVFP4," arXiv preprint arXiv:2509.25149, 2025*
> *[2] Red Hat, "Accelerating large language models with NVFP4 quantization," Red Hat Blog, 2026.*
> *[3] Chen et al. "A statistical framework ...," NeurIPS 2020.*

---

> > ### Author Rebuttal · Reviewer_Lwpw · 2026-04-03
> >
> > The authors’ rebuttal addresses my concerns.

---

> > > ### Author Response · Authors · 2026-04-03
> > >
> > > Dear Reviewer Lwpw,
> > >
> > > Thank you for taking the time to review our rebuttal and for acknowledging that your concerns have been addressed.
> > >
> > > We appreciate your constructive comments during the review process. We will include the additional experiments and discussions from the rebuttal in the revised manuscript.
> > >
> > > Thank you again for your time and effort in reviewing our work.
> > >
> > > Best regards,
> > > The Authors

---

### Official Review · Reviewer_5ZhH · 2026-03-13

**Soundness:** 3
**Presentation:** 3
**Significance:** 3
**Originality:** 3
**Overall Recommendation:** 5
**Confidence:** 2

**Summary:**

This paper describes TetraJet-V2, an effective end-to-end NVFP4 pre-training recipe for LLM. Specfically, the author introduces the following techiques in their implementation: (1) they introduce unbiased double-block quantization to tackle the numerical range limitation of NVFP4, (2) they introduce OsciReset to bound the stability in low-precision weight, (3) they introduce OutControl to minimize the outliers' impact in forward and backward pass activation and gradient.

**Compliance With Llm Reviewing Policy:**

Affirmed.

**Final Justification:**

The author's discussion has resolved my concern regarding their paper. The author had also mostly resolved the other reviewer's concern. Therefore, I decided to recommend the paper more strongly by raising my score from 4 to 5.

**Key Questions For Authors:**

1. In unbiased quantization, the scaling factor is first extend to [-448x6, 448x6] before the outer block is used. Consider there is an additional scaling block, is the inital scaling necessary in this setting, and if so, how is the scale of 6 chosen? Is it an arbitary selection or the choice is backed by ablation?

2. For OsciReset, is the selection of risk threashold $\tau_{osci}$ coupled with $T_{period}$ and $T_{accu}$? Can the author provide some emperical evaluation for how to select this hyperparameter and if the choice of this has significant impact on OsciReset's capability in reducing weight oscilation?

3. In OutControl, the author suggests that RHT is applied to dX and dW because their experiment identified the benefit of RHT in both component. Could the author discuss further on some potential cause on why this differs from the previous finding from Nvidia's recipe.

**Limitations:**

yes

**Strengths And Weaknesses:**

## Strength
* The paper is well structure and the author presents the details for implementation reasonably well.
* The implementation is backed by rigrous experiment to support the claim in the paper.

## Weakness
* It seems like this work is based on TetraJet which is originally aim for tackling quantization pre-training for vision transformer. A more detailed comparison highlighting the difference in the recipe would better illustrate the contribution in this paper.
* The ablation on speedup could be extend to more BS and SeqLen for better reflection of kernel speedup in different experiment setting.

---

> ### Author Rebuttal · Authors · 2026-03-30
>
> We greatly thank the reviewer for valuable comments. We respond to the questions below.
>
> > Weakness 1 (W1):
> > ... needing a more detailed comparison highlighting the difference to TetraJet for Vision Transformers ...
>
> Thanks for the valuable suggestion. We summarize the key differences and our novel contributions compared to the previous TetraJet as follows:
>
> 1. **A novel solution to weight oscillation in LLMs (OsciReset).**
>    Our **core algorithmic innovation** is **OsciReset**, the **first** method to effectively address the weight-oscillation problem specifically in the low-precision training of LLMs. This addresses a unique optimization challenge in low-precision training that is completely orthogonal to outlier control (as shown in paper Sec. 5.2.2) and is largely overlooked in existing literature. Furthermore, our method is simple, with minimal overhead, and requires no intricate hyperparameter tuning (see response to Q2).
>
> 2. **The currently optimal fully-FP4 recipe specifically tailored for LLM training.**
>    While TetraJet focused on ViTs, we successfully identify the optimal fully-FP4 configuration combined with RHT for the LLM training scenario. Finding this optimal recipe is highly non-trivial. Through rigorous theoretical derivation and extensive ablations, we design an effective unbiased backward estimation and optimal RHT settings.
>
>    Applying the recipe correctly in this specific scenario yields large improvements. By simply implementing the correct backward unbiased estimation and suitable RHT settings (*before adding mix-precision or OsciReset*), our **TetraJet-v2-base** reduces the Val. PPL of the 370M model from 26.2 (NVIDIA's method) to 25.5 (closes the PPL-gap-to-BF16 by 28%).
>
> > W2:
> > The ablation on speedup could be extend to more BS and SeqLen.
>
> Thanks for the valuable suggestion. We provide additional end-to-end speedup results for a full Transformer block in *Table 1*. Our accuracy-improved methods **maintain consistent speedup** across different BatchSize and SeqLen.
>
> *Table 1: End-to-End Speedup against TransformerEngine-FP8 (Hidden=8192) on RTX5090*
> |BS|2|2|4|4|8|8|
> |:-|-|-|-|-|-|-|
> | **SeqLen**| 2048 | 4096 | 2048 | 4096 | 2048 | 4096 |
> | TetraJet-v2-base | 1.72x | 1.56x | 1.60x | 1.49x | 1.53x | 1.50x |
> | TetraJet-v2-full | 1.60x | 1.45x | 1.50x | 1.38x | 1.42x | 1.40x |
>
> > Question 1 (Q1):
> > Consider there is an additional scaling block, is the initial scaling necessary in this setting, and if so, how is the scale of 6 chosen? Is it an arbitary selection or the choice is backed by ablation?
>
> Thanks for the valuable question. The choice of "6" and the double-scaling design are strictly determined by **the NVFP4 hardware specification, not by arbitrary selection**.
>
> In NVFP4 format, the inner elements use E2M1 quantization. According to the NVIDIA FP4 standard [1], **the maximum representable value of E2M1 is 6**.
>
> The outer-scaling block is for the NVFP4's E4M3 scaling factor $S_{block}$, since $S_{block}$ can only represent $[-448,448]$ range. After the outer-block scaling, each element is in $[-6\times448, 6\times 448]$, and the initial scaling is still necessary because we need to scale them down to the $[-6, 6]$ range to determine each FP4 value.
>
> > Q2:
> > For OsciReset, is the selection of risk threashold $\tau_{\mathrm{osci}}$ coupled with $T_{\mathrm{period}}$ and  $T_{\mathrm{accu}}$ ? How to select this hyperparameter and if the choice of this has significant impact on OsciReset's capability in reducing weight oscilation?
>
> Thanks for the valuable question. The selection of the risk threshold $\tau_{\mathrm{osci}}$ is **decoupled** from $T_{\mathrm{period}}$ and $T_{\mathrm{accu}}$.
>
> Specifically, we uniformly **fixed $\boldsymbol{\tau_{\mathrm{osci}}=8}$ across all model sizes (70M to 370M) and data scales** in our experiments. This consistency demonstrates the strong scalability and robustness of this empirical choice.
>
> To further address the concern, we conducted an additional ablation study on varying $\tau_{\mathrm{osci}}$. As shown in the table below, **the final performance is not sensitive to the exact value of $\tau_{\mathrm{osci}}$** and OsciReset consistently brings significant improvements over the baseline.
>
> *Table 2: Insensitivity to $\tau_{\mathrm{osci}}$ on OLMo-2-150M with 107B tokens training.*
> | $\tau_{\mathrm{osci}}$ | Baseline (w/o OsciReset) | 4| 8| 12| 16|
> |-|-|-|-|-|-|
> | Val. PPL| 35.54| 34.99 | **34.95** | 35.00 | 35.01 |
>
> > Q3:
> > ... why the selection of RHT differs from the previous finding from Nvidia's recipe?
>
> The primary cause is that NVIDIA's recipe **misaligns** the activation between the forward and backward passes, which compromises gradient correctness. Once we corrected this alignment to ensure strictly **unbiased** gradient estimation, we found that applying RHT to both dW and dX computations becomes consistently beneficial.
>
> *[1] NVIDIA, "Pretraining Large Language Models with NVFP4", arXiv preprint arXiv:2509.25149, 2025*

---

> > ### Author Rebuttal · Reviewer_5ZhH · 2026-04-04
> >
> > I thank the authors for further clarification on the topic. I would suggest that the authors incorporate the additional discussion with W1 and W2 into the revised version of their paper.

---

> > > ### Author Response · Authors · 2026-04-04
> > >
> > > Dear Reviewer 5ZhH,
> > >
> > > Thank you for reviewing our rebuttal and confirming that your concerns have been fully resolved. We truly appreciate your valuable suggestions.
> > >
> > > Following your advice, we will explicitly incorporate the discussions on W1 and the extended speedup results in W2 into the revised manuscript. We believe these additional details will naturally enrich our text and help make our original contributions clearer to the readers.
> > >
> > > Thank you again for your time and constructive feedback.
> > >
> > > Best regards,
> > > The Authors

---

### Decision · Program_Chairs · 2026-04-30

**Decision:**

Accept (spotlight)

**Comment:**

This paper proposes a fully-quantized training method for LLMs that addresses the key bottlenecks of oscillation and outliers. While the reviewers raised some concerns, the author managed to resolve them during rebuttal, with all reviewers leaning towards clear accept. Therefore, I'd recommend acceptance.